# MutualVPR: A Mutual Learning Framework for Resolving Supervision Inconsistencies via Adaptive Clustering

Qiwen Gu[1]    Xufei Wang[3]    Junqiao Zhao[1,2] *    Siyue Tao[1]    Tiantian Feng[4]
Ziqiao Wang[1]    Guang Chen[1,5]

[1]School of Computer Science and Technology, Tongji University, Shanghai, China
[2]MOE Key Lab of Embedded System and Service Computing, Tongji University, Shanghai, China
[3]Shanghai Research Institute for Intelligent Autonomous System, Tongji University, Shanghai, China
[4]School of Surveying and Geo-Informatics, Tongji University, Shanghai, China
[5]Shanghai Innovation Institute
{2432178, tjwangxufei, zhaojunqiao, 2331923, Fengtiantian}@tongji.edu.cn
{ziqiaowang, guangchen}@tongji.edu.cn

## Abstract

Visual Place Recognition (VPR) enables robust localization through image retrieval based on learned descriptors. However, drastic appearance variations of images at the same place caused by viewpoint changes can lead to inconsistent supervision signals, thereby degrading descriptor learning. Existing methods either rely on manually defined cropping rules or labeled data for view differentiation, but they suffer from two major limitations: (1) reliance on labels or handcrafted rules restricts generalization capability; (2) even within the same view direction, occlusions can introduce feature ambiguity. To address these issues, we propose MutualVPR, a mutual learning framework that integrates unsupervised view self-classification and descriptor learning. We first group images by geographic coordinates, then iteratively refine the clusters using K-means to dynamically assign place categories without orientation labels. Specifically, we adopt a DINOv2-based encoder to initialize the clustering. During training, the encoder and clustering co-evolve, progressively separating drastic appearance variations of the same place and enabling consistent supervision. Furthermore, we find that capturing fine-grained image differences at a place enhances robustness. Experiments demonstrate that MutualVPR achieves state-of-the-art (SOTA) performance across multiple datasets, validating the effectiveness of our framework in improving view direction generalization, occlusion robustness. The code can be found at `https://github.com/Gucci233/MutualVPR`.

## 1 Introduction

Visual Place Recognition (VPR) is the task of determining a previously visited location from a query image by matching it against a database of geo-tagged reference images. It serves as a key component in long-term localization and loop closure for autonomous systems such as mobile robots [10, 11, 32] and self-driving vehicles [12, 17].

Existing VPR methods either utilizes contrastive learning [4, 30, 20, 15, 2] or classification-based learning [26, 22, 6, 7] to learn the place representation. Contrastive learning-based approaches

---

*Corresponding author

39th Conference on Neural Information Processing Systems (NeurIPS 2025).

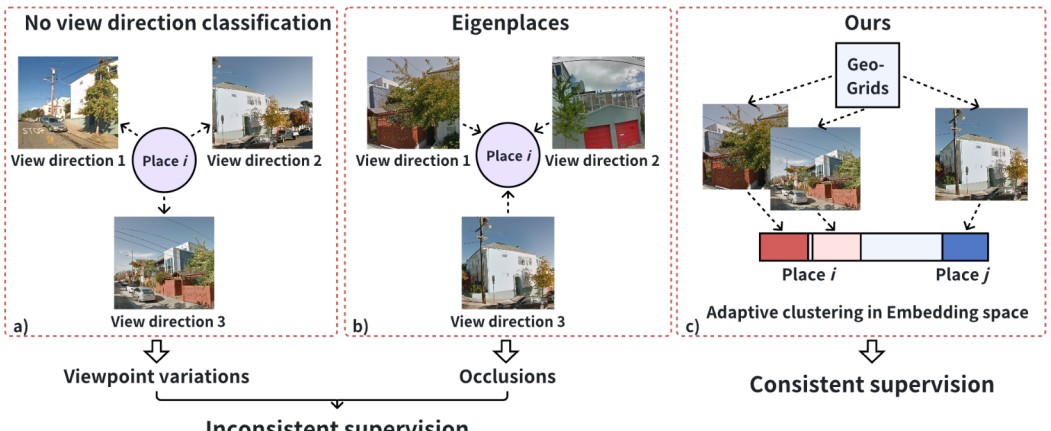

Figure 1: **The problem of inconsistent supervision in existing VPR researches.** The proposed mutual-learning framework define place labels by adaptive clustering in embedding space, enforcing supervision consistency.

facilitate the learning of robust and discriminative descriptors. However, they depend heavily on hard sample mining, which incurs significant computational cost and limits scalability to large-scale. Classification-based VPR methods divide the environment into spatial grids based on geographic coordinates, assigning each grid cell a unique class label, therefore, avoiding the need for expensive sample mining.

However, as shown in Figure 1 a), views captured from the same location but in different directions can result in drastically different visual scenes. If view direction is not considered, such views are treated as originating from the same place, which may introduce inconsistent supervision signals.

CosPlace [6] first attempts to address this problem by manually defining class labels based on view directions. However, this method does not provide explicit guarantees of intra-class visual similarity or inter-class visual distinctiveness. As a result, view variation may still persist. EigenPlaces [7] proposes a cropping strategy with the assumption that views toward a common reference point are similar. However, as shown in Figure 1 b), real-world scenes often involve occlusions from buildings, vehicles, and other structures. Such occlusions can lead to significant visual differences thus introduce inconsistent supervision. Consequently, visually dissimilar scenes may be incorrectly grouped under the same label. Such supervision inconsistency misaligns the supervisory signal with the true visual similarity, thereby undermining the model's ability to learn robust and discriminative features.

To address the problem of supervision inconsistency, we propose MutualVPR, a mutual learning framework that jointly refines image descriptors from the same geo-grids and view classification as shown in Figure 1 c). Unlike prior methods that rely on fixed or heuristically defined place labels, MutualVPR dynamically updates both feature representations and place label assignments through iterative, feature-driven mutual learning. This co-evolution enables the system to align semantic content with supervision more effectively.

Our main contributions can be summarized as follows:

- We propose a mutual learning framework where feature learning and clustering co-evolve, effectively mitigating supervision inconsistency.

- A adaptive clustering strategy dynamically refines pseudo-labels based on visual semantics, handling view directions and occlusions without orientation labels.

- Our method achieves state-of-the-art (SOTA) performance on challenging VPR benchmarks, demonstrating robustness to diverse appearance variations and inconsistent supervision.

## 2 Related Works

### 2.1 Contrastive Learning-based VPR

Recent advances in VPR have largely benefited from deep learning-based approaches. Methods such as NetVLAD [4] pioneered the use of trainable aggregation layers to produce compact and discriminative global image descriptors. Contrastive learning [8, 24] has become prevalent, employing triplet-based or contrastive objectives to encourage the model to learn discriminative representations.

However, images taken at the same location may be captured from very different view directions—resulting in large visual appearance gaps between positive samples. Conversely, images from nearby but distinct locations may look similar due to aligned view directions, increasing the risk of erroneous negatives. This mismatch between place labels and visual similarity can mislead contrastive objectives and degrade performance.

To mitigate the impact of view direction variation, MixVPR [2], CricaVPR [20], SALAD [15] and BoQ [3] train on the GSV-Cities dataset [1], where all images within the same class share a consistent view direction. By ensuring this, these methods reduce inconsistent supervision.

Sample mining strategies such as GCL [19] and Clique Mining (CM) [14] further enhance learning: GCL assigns graded similarity labels to reduce supervision noise, while CM forms batches of very similar images to create harder training samples.

Despite these improvements, all these approaches still face challenges when images from the same location exhibit diverse view directions or occlusions, limiting their generalization under extreme conditions.

Other methods [9, 13, 33, 21] aim to enhance robustness by incorporating local feature matching, but they often rely on a two-stage process, which incurs significant computational overhead.

### 2.2 Classification-based VPR

Despite their [2, 20, 15, 3, 18] success, contrastive learning-based methods rely on hard sample mining, which increases training complexity and computational overhead. An alternative approach is to formulate VPR as a classification problem, reducing the need for explicit pairwise comparisons. Methods such as DaC [28] directly partitions images into grids based on their geographic coordinates, training the feature extractor by classifying images into their corresponding grids. However, it does not take into account the variations in viewpoint within each grid. Furthermore, CosPlace [6] and EigenPlaces [7] categorize images into location and view-based classes, allowing efficient training with categorical cross-entropy loss.

CosPlace[6] partitions the dataset into geographic cells and classifies images based on view direction labels. However, this rule-based scheme overlooks visual similarity among samples within the same cell. As a result, it often assigns semantically similar images to different class and vice versa, leading to inconsistent supervision.

EigenPlaces [7] introduces a classification scheme based on the Singular Value Decomposition (SVD) of image locations, grouping images that share a common reference point. A key advantage of this approach is its independence from manually defined view direction labels. However, the underlying assumption—that images oriented toward the same reference point share similar visual content—often fails in urban environments due to occlusions caused by buildings, vehicles, or vegetation. Consequently, such scenes may exhibit substantial visual differences despite similar viewing intent, resulting in supervision inconsistencies that undermine classification reliability.

## 3 Problem Analysis

To better understand the issue of supervision inconsistency in classification-based VPR methods, we visualize the image descriptors extracted by CosPlace [6], EigenPlace [7], and our method using t-SNE.

As shown in Figure 2, each View corresponds to an image captured at a specific geographic position and camera orientation, while a Class denotes a group of views considered to represent the same place.

In practice, class labels can be assigned based on orientation labels (as in CosPlace), where each direction corresponds to a predefined class. However, this scheme often fails to align with true scene semantics, leading to inconsistent supervision when visually similar views fall into different classes. Alternatively, labels derived from descriptor clustering group images by visual similarity, naturally ensuring semantic consistency.

Such supervision inconsistencies mainly manifest as view variations and occlusions. View variation can be further divided into two types:

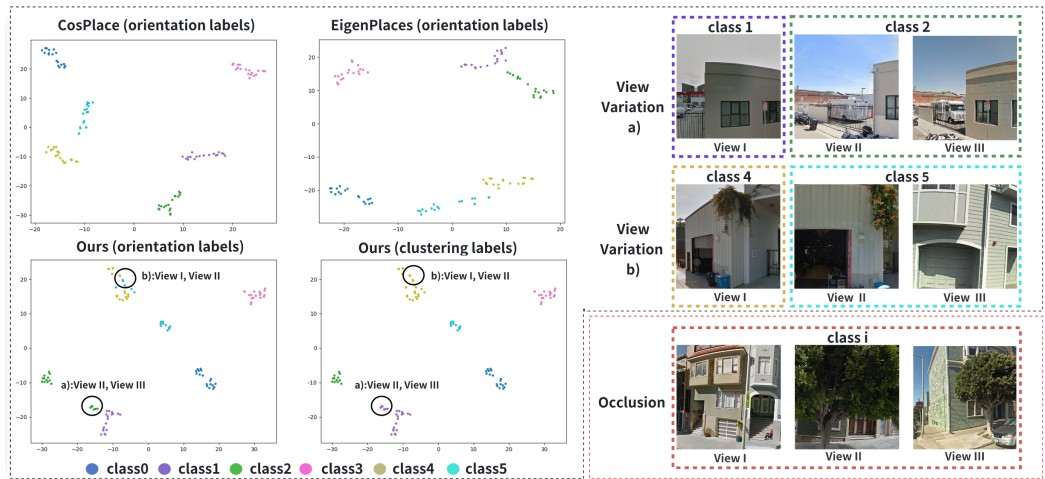

Figure 2: **Supervision Inconsistency in Classification-based Methods.** The left panel shows t-SNE visualizations of image descriptors extracted from a single geo-grid using different methods on the SF-XL dataset. "orientation labels" indicate samples colored according to their assigned view directions, while "clustering labels" refers to labels obtained by applying K-means clustering to the descriptors. The top-right panel illustrates issues related to view variation, using image examples drawn from samples in the t-SNE visualization on the left. The bottom-right panel illustrates occlusion induced issues, using image example from the same reference point.

**View variation a):** Views with large scene overlap are assigned to different labels due to orientation labels based on view direction. In panel a), Views II and III are labeled as Class 2, while the highly similar View I is labeled as Class 1. As shown in the t-SNE plots of CosPlace and EigenPlaces, their features directly inherit this flawed supervision, forming two separate clusters despite the strong visual overlap. When our learned features are colored by orientation labels, this artificial split remains; however, coloring the same features by clustering results yields a single, coherent group. This contrast highlights the core issue — the conflict between fixed directional supervision and actual visual semantics.

**View variation b):** The opposite inconsistency occurs when visually distinct views are assigned the same label. In panel b), Views II and III share Class 5 despite clear visual differences, while View I (Class 4) is semantically closer to View II. Feature visualizations colored by orientation labels reflect this mistaken grouping, whereas clustering-based coloring reorganizes them into more meaningful, semantically consistent clusters. This again exposes the mismatch between visual reality and rigid label assignment, underscoring the need for adaptive supervision.

Occlusion-induced inconsistencies are demonstrated by "Occlusion" in Figure 2, where images captured from different viewpoints but oriented toward the same reference point may exhibit significantly different visual content due to obstacles. In the "Occlusion" panel, Views I, II and III are captured from the same geo-grid but are occluded by trees or different buildings, leading to distinct visual content. Although these views share the same reference point, their semantic content diverges. This violates the assumptions of EigenPlace, resulting in supervision inconsistency.

We therefore argue that effective supervision for VPR should reflect semantic similarity rather than rely solely on spatial proximity or fixed directional assumptions. Without semantically consistent supervision, models are prone to learning unstable features, reducing their ability to generalize across complex, realworld scenarios.

# 4 Approach

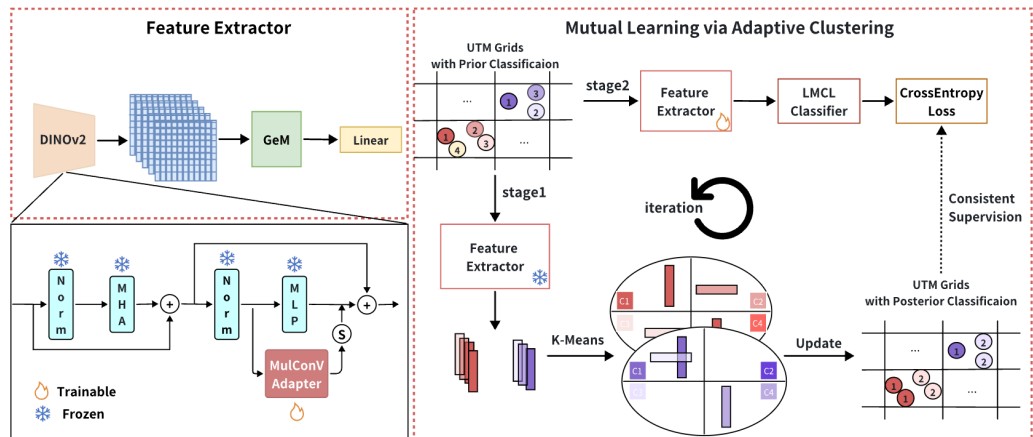

Figure 3: **Mutual Learning Framework via Adaptive Clustering.** We initialize spatial grids using UTM coordinates and assign coarse intra-grid categories. Features are extracted using DINOv2[23] with adapter and GeM [25] pooling, while adaptive clustering where iterative K-means is guided by LMCL loss dynamically refines view direction categories within grids. Clusters and features co-evolve: updated clusters supervise feature learning (stage 1), and improved features guide reclustering (stage 2), enabling robust supervision under occlusions and view direction changes.

The proposed MutualVPR framework integrates unsupervised view self-classification with joint descriptor training in a mutual learning paradigm (as shown in Figure 3). The key lies in its adaptive clustering mechanism, which iteratively refines place categories throughout training.

## 4.1 Feature Encoder

MutualVPR is built upon DINOv2 [23], and incorporates the MulConv adapter [20] to enhance the model's capacity for robust feature representation. MulConv adopts a bottleneck structure with three parallel convolutional branches operating at different receptive fields, enabling the extraction of multi-scale features. This design improves the model's ability to handle variations in object scale and spatial context, which is beneficial in environments with diverse structural patterns. It can be expressed as:

$$
\begin{aligned}
z_l' &= \text{MHA}\left(\text{LN}\left(z_{l-1}\right)\right) + z_{l-1}, \\
z_l &= \text{MLP}\left(\text{LN}\left(z_l'\right)\right) + s \cdot \text{Adapter}\left(\text{LN}\left(z_l'\right)\right) + z_l'.
\end{aligned}
\tag{1}
$$

where $z_{l-1}$ denotes the input features from the previous layer, $\text{LN}(\cdot)$ is Layer Normalization, and $\text{MHA}(\cdot)$ refers to Multi-Head Attention. $\text{MLP}(\cdot)$ stands for the feed-forward network, while $\text{Adapter}(\cdot)$ represents the MulConv adapter module. The parameter $s$ is a learnable scaling factor that controls the contribution of the adapter branch.

## 4.2 Mutual Learning via Adaptive Clustering

### 4.2.1 Place Label Initialization

Similar to CosPlace[6], but without using view direction labels, we first partition images into coarse location-based classes using UTM coordinates. At this stage, view direction variations within the same location are not considered. Formally, a coarse location class is defined as:

$$
x = \left\{ \left\lfloor \tfrac{east}{M} \right\rfloor = e_i, \left\lfloor \tfrac{north}{M} \right\rfloor = n_j \right\},
\tag{2}
$$

where $M$ is a hyperparameter controlling the size of the grid. This step serves as an initialization of place labels, which are later refined through adaptive clustering.

#### 4.2.2 Mutual Learning of Feature Extraction and Clustering

While UTM-based grouping ensures spatial locality, it fails to capture appearance variations caused by view direction changes and occlusions. To address this, we introduce a mutual learning framework that combines feature-aware supervision with adaptive clustering.

Within each coarse UTM region, we perform iterative K-means clustering based on learned descriptors, enabling view direction categories to adapt dynamically as feature representations evolve. Formally, view direction categories are defined as:

$$C = \{e_i, n_j, h \mid e_i, n_j \in x, h \in K\}, \tag{3}$$

where $K$ controls the granularity of view direction partitioning. Unlike static classification methods with fixed view direction labels, our approach allows continuous refinement of view direction clusters, better aligning them with actual visual similarity.

As training progresses, the feature encoder and clustering process update each other iteratively. This self-correcting mechanism avoids error accumulation from early misassignments and ensures images are grouped into more consistent view direction categories over time.

Following CosPlace [6], we adopt Large Margin Cosine Loss (LMCL) [29] as our classifier to enforce discriminative feature learning. At inference time, descriptors are directly extracted from the feature encoder and used for image retrieval based on their distances.

### 4.3 Implementation Details

In the implementation, DINOv2's ViT-B 14 is employed as the backbone network, with all parameters frozen except for the Adapter. Input image size is resized to 504×504 to meet the input requirements of ViT-B 14. GeM pooling and a fully connected layer reduce the dimensionality to 512 for the final descriptor. For dataset classification, margin $M$ is set to 10, and $K$ clusters are set to 3. The feature extractor is initialized with a learning rate of 1e-5, while the classifier uses 1e-2. We employ the Adam optimizer and apply a cosine annealing scheduler to the feature extractor. Training is conducted for 50 epochs, each consisting of 10,000 iterations. To reduce training time due to the huge dataset, we divide all UTM classes from the coarse classification into eight groups, with each group serving as the training set for one epoch. In each epoch, one-fifth of the classes within the selected group is randomly chosen for feature-aware clustering.

During training, we found that increasing overlapping ratio of cropped images can enhance semantic continuity. For the multi-angle cropping strategy, we crop panoramic images from different starting angles every 60° to generate training data. In our experiments, the starting angles are set to 0° and 30°. Since all images are cropped from panoramas, adjacent images naturally share semantic content. In our work, adjacent classes—such as class 2 and 3—represent neighboring view directions.

## 5 Experimental Results

### 5.1 Research Questions

In this work, we focus on the supervision inconsistency problem in VPR. We aim to investigate the following research questions:

**Q1**: How does our method perform compared to SOTA VPR approaches across standard benchmarks?

**Q2**: How well does our method generalize to challenging dataset with occlusion?

**Q3**: How does our adaptive clustering compare to orientation labels in handling supervision inconsistency?

### 5.2 Datasets and Evaluation Metrics

For contrastive learning-based VPR methods, we use the GSV-Cities dataset [1] for training. For classification-based VPR methods, we use the SF-XL dataset [6] for training. The selected training set is a subset of about 0.9M panoramic images, following CosPlace. A multi-angle cropping strategy as describe in Section 4.3 is applied. Eigenplaces use all panoramic images of about 3.4M for cropping.

For evaluation, we adopt several widely used VPR benchmarks: Pitts30k-test, Pitts250k-test [4], MSLS-val [31], Tokyo 24/7 [27], and SF-XL-test v1 [6]. To assess the generalization capability of our method, we also evaluate it on SF-XL-occlusion [5] dataset. This dataset, originally designed to assess the robustness of VPR methods under severe occlusion, also exposes the limitations of conventional supervision strategies. The detailed statistics of the datasets can be seen in Appendix A.

We employ the standard Recall@K metric, defined as the ratio of correctly located queries to the total number of queries. Correct localization involves searching for positive examples by matching images within 25-meter radius threshold based on geographical coordinates.

The experiment is executed on a server with three NVIDIA RTX 3090 GPUs, using PyTorch for training and testing.

## 5.3 Comparison with Other Methods

| Method | Desc.dim. | Train set | MSLS-val | | Pitts30k | | Pitts250k | | Tokyo24/7 | | SF-XL-testv1 | |
|---|---|---|---|---|---|---|---|---|---|---|---|---|
| | | | R@1 | R@5 | R@1 | R@5 | R@1 | R@5 | R@1 | R@5 | R@1 | R@5 |
| NetVLAD | 131072 | GSV-Cities | 82.8 | 90.3 | 87.0 | 94.3 | 89.1 | 94.8 | 69.5 | 82.5 | - | - |
| GeM | 2048 | GSV-Cities | 72.5 | 82.7 | 84.5 | 92.8 | 85.1 | 93.4 | 60.3 | 73.7 | 25.6 | 35.5 |
| AnyLoc(ViT-B+GeM) | 768 | - | 32.6 | 41.6 | 77.7 | 88.9 | 79.3 | 89.5 | 71.7 | 87.6 | 33.3 | 45.2 |
| ConvAP | 2048 | GSV-Cities | 81.5 | 87.5 | 89.7 | 95.2 | 91.2 | 96.4 | 74.6 | 83.2 | 41.1 | 53.0 |
| MixVPR | 4096 | GSV-Cities | 87.1 | 91.4 | 91.6 | 95.5 | **94.3** | 98.1 | 87.0 | 93.3 | 69.2 | 77.4 |
| CricaVPR* | 4096 | GSV-Cities | 90.0 | 95.4 | 94.9 | 97.3 | – | – | 93.0 | 97.5 | - | - |
| CricaVPR$_1$ | 4096 | GSV-Cities | 88.5 | 95.1 | 91.6 | 95.7 | **94.3** | **98.6** | 89.5 | 94.6 | 72.8 | 80.1 |
| CricaVPR$_1$ (PCA) | 512 | GSV-Cities | 87.1 | 92.6 | 90.4 | 94.9 | 92.5 | 97.1 | 87.4 | 92.9 | 68.4 | 77.1 |
| BoQ$^\dagger$ | 512 | GSV-Cities | 88.4 | 93.9 | **93.1** | 96.1 | 93.8 | 97.5 | 91.9 | 95.5 | 79.6 | 85.9 |
| SALAD$^\dagger$ | 512+32 | GSV-Cities | 88.5 | 94.2 | 90.6 | 95.1 | 92.1 | 97.0 | 92.3 | 95.1 | 70.2 | 77.7 |
| SALAD+CM$^\dagger$ | 512+32 | MSLS+GSV-Cities | **90.4** | **96.2** | 90.9 | 95.9 | 93.2 | 97.8 | **92.8** | 96.2 | 78.4 | 85.4 |
| EigenPlaces | 512 | SF-XL | 88.1 | 92.9 | 92.3 | 96.1 | 93.5 | 97.5 | 84.8 | 94.0 | **83.8** | **89.6** |
| CosPlace | 512 | SF-XL | 84.4 | 90.2 | 89.6 | 94.9 | 90.4 | 96.6 | 76.5 | 89.2 | 64.8 | 73.1 |
| MutualVPR (Ours) | 512 | SF-XL | 89.2 | 95.1 | 90.9 | **96.4** | 92.6 | 97.9 | 92.4 | **96.6** | 80.8 | 86.4 |

Table 1: **Comparison of various methods on multiple benchmark datasets.** The upper block lists contrastive learning methods, while the lower block lists classification-based methods. CricaVPR* denotes the results from the original paper relying on batch interaction, and CricaVPR$_1$ is our single-query variant. $^\dagger$ indicates a retrained 512-D or 512+32-D version. Best results are shown in **bold**, second best are underlined.

To answer the research question Q1, we compare our method with SOTA VPR methods, including both contrastive learning-based and classification-based approaches. The former includes NetVLAD [4], GeM [25], AnyLoc [18], ConvAP [1],MixVPR [2], CricaVPR [20], BoQ [3], SALAD [15] and CM [14]. The latter includes EigenPlaces [7], and CosPlace [6].

It should be noted that our NetVLAD is trained on a ResNet-50 backbone, unlike the original version trained on VGG16. Nevertheless, its descriptor dimensionality is still very high, which prevents evaluation on the SF-XL-test set due to memory constraints when processing 2M samples. The comparison results are shown in Table 1. It shows that MutualVPR consistently delivers competitive or superior performance across multiple benchmarks, showing strong generalization capability.

For classification-based baselines, MutualVPR generally outperforms CosPlace across datasets, even when CosPlace uses ground-truth labels. This demonstrates the benefit of our adaptive clustering, which mitigates the limitations of orientation labels with fixed splits that may misrepresent visual similarity under viewpoint changes or occlusions. EigenPlaces, on the other hand, is specifically designed to handle extreme viewpoint variations. While it performs well on SF-XL-testv1, its performance drops on more diverse datasets such as Tokyo 24/7 and MSLS-Val. This is because EigenPlaces' strong focus on viewpoint invariance can limit its generalization to scenarios involving occlusions or other challenging conditions. In contrast, MutualVPR achieves a balance: it effectively handles viewpoint variations while remaining robust to occlusions and other extreme conditions, leading to better overall generalization. Supporting experiments validate this claim in Appendix D.

Among contrastive learning-based methods, those trained on the GSV-Cities dataset generally exhibit strong performance. This is largely attributed to the nature of GSV-Cities, which contains multiple captures of the same location under highly consistent viewpoints. Such data inherently mitigates supervision inconsistencies caused by view variations, allowing contrastive methods to learn more stable representations. However, achieving this level of performance requires large-scale, carefully

curated data and extensive sample mining. Recent state-of-the-art methods, such as SALAD+CM, further leverage large and diverse training sets (MSLS + GSV-Cities) to achieve the highest average performance across multiple benchmarks.

In contrast, our MutualVPR is trained solely on SF-XL yet achieves robust and well-balanced performance across diverse conditions. Although not always the top performer on every individual benchmark, it ranks closely behind SALAD+CM in overall accuracy and even surpasses it on Pitts30k and Tokyo24/7 in terms of R@5. Moreover, unlike methods such as CricaVPR*, which rely on batch-level feature interaction to enhance localization accuracy and suffer a substantial drop in single-query inference (as indicated by CricaVPR$_1$), our approach maintains stable performance without requiring inter-sample dependencies, further underscoring its practical robustness.

### 5.4 Evaluate on Occlusion Dataset

To answer research questions Q2, we evaluate the generalization capability of our method using a dataset characterized by significant occlusions, which further reflects the impact of inconsistent supervision on retrieval performance. Table 2 presents the retrieval performance on the SF-XL-Occlusion dataset, where each query is affected by occlusion.

Our method achieves 47.4% R@1, significantly outperforming all baselines, including EigenPlaces (36.8%) and CosPlace (32.9%), as well as several contrastive learning methods such as MixVPR (30.3%) and SALAD (31.6%). Among the contrastive learning baselines, CricaVPR$_1$ (40.8%), SALAD+CM (40.8%), and BoQ (38.2%) show competitive performance, but still remain below our method for top-k retrieval metrics. These results demonstrate that our approach maintains robust retrieval under heavy occlusion.

| Method | Desc.dim. | SF-XL-Occlusion | | | |
| --- | --- | --- | --- | --- | --- |
| | | R@1 | R@5 | R@10 | R@20 |
| GeM | 2048 | 11.8 | 15.8 | 17.1 | 22.4 |
| AnyLoc(ViT-B+GeM) | 768 | 6.6 | 14.5 | 19.7 | 26.3 |
| ConvAP | 2048 | 23.7 | 26.3 | 28.9 | 31.6 |
| MixVPR | 4096 | 30.3 | 35.5 | 38.2 | 44.7 |
| CricaVPR$_1$ | 4096 | 40.8 | 51.3 | 54.6 | 59.9 |
| BoQ$^\dagger$ | 512 | 38.2 | 50.0 | 53.3 | 59.2 |
| SALAD$^\dagger$ | 512+32 | 31.6 | 42.1 | 46.1 | 51.3 |
| SALAD+CM$^\dagger$ | 512+32 | 40.8 | 53.7 | 58.3 | 61.3 |
| EigenPlaces | 512 | 36.8 | 51.8 | 56.6 | 59.2 |
| CosPlace | 512 | 32.9 | 43.4 | 46.1 | 48.7 |
| No Classification | 512 | 17.1 | 25.0 | 26.3 | 31.6 |
| **MutualVPR (Ours)** | 512 | **47.4** | **65.8** | **71.1** | **73.7** |

Table 2: **Comparison on SF-XL-Occlusion.** Each query in SF-XL-Occlusion is affected by occlusion, making it suitable for testing robustness under missing visual cues. The upper block lists contrastive learning methods, and the lower block lists classification-based ones. "No Classification" indicates that no intra-grid classification is applied—images within the same grid are treated as belonging to the same class. Best results are shown in **bold**, and second best are underlined.

Our strong performance under occlusion stems from two key advantages:

**Adaptive Supervision Consistency:** Unlike static supervision methods (e.g. CosPlace, EigenPlaces), our approach refines class assignments through iterative, feature-aware clustering. This process corrects initial supervision errors caused by occlusions or view changes, progressively aligning features from occluded and unoccluded samples belonging to the same place. As a result, the model learns to associate semantically similar scenes across varying view directions, improving label consistency and robustness.

See Appendix B.1 for a visual example showing a misclassified occluded query being reassigned to the correct class after training.

**Semantic Proximity within Class:** Our method brings semantically similar views closer in feature space, even when they originate from different view directions. This contrasts with rigid label-based schemes, where visually similar scenes are separated due to orientation labels.

This semantic proximity benefits retrieval: even if an occluded query is not correctly classified, it can still be retrieved as long as its feature lies within the threshold distance.

We conducted experiments showing that our method achieves consistently lower inter-class distances than CosPlace and EigenPlaces across adjacent view directions. Quantitative analysis and distance comparisons are presented in Appendix B.2.

These findings suggest that by iteratively updating class assignments based on feature similarity, our approach naturally brings semantically similar scenes—despite view differences or occlusions—closer in the feature space. This results in more compact and coherent class boundaries.

In contrast, these methods which rely on rigid, manually defined labels often split visually similar scenes into separate classes, creating artificial gaps in the feature space. Our method mitigates such fragmentation, enabling smoother transitions between adjacent classes and enhancing retrieval robustness.

## 5.5 Ablations

### 5.5.1 Adaptive vs Static Label Supervision

To answer research question Q3, we compare manually assigned view direction labels with our adaptive clustering approach across various settings.

- Fixed view direction Labels (CosPlace) : Using predefined view direction labels from geographic metadata.
- Fixed view direction Labels (CosPlace) + Cropping: Applying multi-angle cropping while still relying on predefined view direction labels.
- Adaptive Clustering (Ours): Using our adaptive clustering but training on raw images without multi-angle cropping.
- Adaptive Clustering (Ours) + Cropping: Incorporating both self-adaptive clustering and multi-angle cropping.

| Method / Diff. | Backbone | Cropping strategy | Tokyo24/7 | | SF-XL-testv1 | |
|---|---|---|---|---|---|---|
| | | | R@1 | R@5 | R@1 | R@5 |
| CosPlace | ResNet50 | 0° | 76.5 | 89.2 | 64.8 | 73.1 |
| CosPlace | ResNet50 | 30° | 80.1 | 90.2 | 70.1 | 80.6 |
| CosPlace | ResNet50 | 0°+30° | 85.1 | 92.4 | 81.1 | 86.2 |
| MutualVPR(Ours) | ResNet50 | 0° | 82.9 | 90.2 | 74.5 | 83.1 |
| MutualVPR(Ours) | ResNet50 | 30° | 81.6 | 91.0 | 72.4 | 81.6 |
| MutualVPR(Ours) | ResNet50 | 0°+30° | 85.4 | 92.5 | 74.8 | 82.5 |
| CosPlace | DINOv2 | 0° | 90.2 | 95.3 | 76.6 | 86.3 |
| CosPlace | DINOv2 | 30° | 89.8 | 95.0 | 75.9 | 86.1 |
| CosPlace | DINOv2 | 0°+30° | 91.0 | 95.8 | 79.1 | 86.3 |
| MutualVPR(Ours) | DINOv2 | 0° | 91.1 | 97.1 | 77.0 | 84.6 |
| MutualVPR(Ours) | DINOv2 | 30° | 89.9 | 96.0 | 78.1 | 84.4 |
| MutualVPR(Ours) | DINOv2 | 0°+30° | 92.1 | 96.5 | 80.8 | 86.4 |

Table 3: **Performance Comparison of Ground Truth and Cropping.** CosPlace represents a method that uses ground-truth and our method represents mutual learning frame. 0° and 30° indicate the starting angles when cropping the panorama.

We conducted experiments using both DINOv2 and ResNet50 as backbones. For each starting angle, we cropped the panoramic image into six evenly spaced views; e.g. with starting angles of 0° and 30°, this results in a total of 12 cropped images. In our method using ResNet50, we clustered all views into six classes ($K = 6$), while for DINOv2, we followed the same procedure but set $K = 3$.

While CosPlace sees clear gains from multi-angle cropping on SF-XL-testv1 (R@1 up to 81.1%), its performance on Tokyo24/7 (R@1 at 85.1%) reveals limited generalization, likely due to overfitting.

In contrast, MutualVPR achieves consistently strong results across both datasets. Both methods benefit from the multi-angle cropping strategy, which increases semantic continuity, but MutualVPR gains more thanks to its adaptive clustering that better aligns semantically similar views.

### 5.5.2 Cluster Number

The number of clusters, $K$, is a hyperparameter that determines the granularity of dataset partitioning using K-means. With fewer clusters, the images within each class tend to be more compact and exhibit higher similarity in the feature space. Conversely, a larger $K$ results in finer partitioning, increasing the diversity of images within the feature space. In our experiments, we evaluated $K = 1, 3, 6$, and the corresponding results are shown in Table 4.

| Backbone | K | Tokyo24/7 | | SF-XL-testv1 | |
|---|---|---|---|---|---|
| | | R@1 | R@5 | R@1 | R@5 |
| ResNet50 | 1 | 68.3 | 84.8 | 52.1 | 62.0 |
| ResNet50 | 3 | 81.0 | 89.8 | 73.9 | 81.4 |
| ResNet50 | 6 | 82.9 | 90.2 | 74.5 | 83.3 |
| DINOv2 | 1 | 80.6 | 91.4 | 61.1 | 71.1 |
| DINOv2 | 3 | 91.1 | 97.1 | 77.0 | 84.6 |
| DINOv2 | 6 | 86.0 | 94.0 | 70.9 | 79.6 |

Table 4: **Different Cluster Numbers.** Trained on the original dataset, results with a descriptor dimension of 512. $K = 1$ can be considered as not classifying the dataset, while $K = 6$ aims to match the number of cluster labels with the ground truth clustering labels.

As expected, the case without clustering ($K = 1$) yields the worst performance, demonstrating the effectiveness of incorporating view direction classification. For the ResNet50 backbone, performance is highest when $K = 6$, though the improvement over $K = 3$ is marginal. In contrast, for the DINOv2 backbone, optimal performance is achieved with $K = 3$, outperforming $K = 6$ by approximately 10%. These results suggest that the optimal number of clusters depends not only on the dataset but also on the backbone. Determining an appropriate $K$ for a given dataset and model remains an open question in the context of the proposed method.

More ablation studies can be found in the Appendix C.

## 6 Conclusion and Future Work

We proposed MutualVPR, a mutual learning framework that addresses supervision inconsistencies in VPR caused by view direction variations and occlusions. By dynamically refining view direction categories through adaptive clustering guided by feature learning, our method eliminates reliance on orientation labels and achieves semantically consistent supervision. Extensive experiments show that MutualVPR achieves robust and generalizable performance across diverse and challenging datasets, validating the effectiveness of our adaptive clustering strategy for real-world VPR applications.

A limitation of our approach is the fixed cluster number $K$, which may not fully capture varying view direction distributions. Since classification and descriptor learning are mutually reinforced, a static $K$ may limit model's adaptability. Our studies underscore its impact on performance, suggesting the need for dynamic adjustment. Future work will explore adaptive clustering to optimize $K$ based on dataset characteristics, enhancing the synergy between classification and representation learning.

## Acknowledgments and Disclosure of Funding

This work is supported by the National Key Research and Development Program of China (No. 2020YFA0711402).

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

## A  Dataset Details

We train on several large-scale datasets, including SF-XL[6] for classification-based methods , GSV-Cities [1] for contrastive learning, and evaluate on Pitts30k-test [4], MSLS-val [31], Tokyo 24/7 [27].

GSV-Cities is a large-scale dataset containing 560k images depicting 67k unique places captured from consistent viewpoints, each labeled with geographic coordinates. SF-XL consists of images cropped from panoramic views at the same location, covering diverse viewing angles and acquisition years, making it suitable for learning viewpoint- and time-robust representations.

A summary of above datasets is provided in Table 5.

| Dataset | Database | Query |
|---|---|---|
| SF-XL-train | 5.6M | |
| SF-XL-test | 2M | 1000 |
| SF-XL-val | 8K | 8064 |
| SF-XL-occlusion | 2M | 76 |
| GSV-Cities-train | 560K | |
| Pitts30k-test | 10K | 6818 |
| MSLS-val | 18.9k | 740 |
| Tokyo24/7-test | 76K | 315 |

Table 5: **Experimental dataset statistics.**

## B  Supervision Correction and Feature Distance Analysis

To further support our claims on supervision correction and semantic proximity, we provide additional visualizations and feature-level analysis of our method.

### B.1  Visualization of Supervision Correction

To demonstrate how adaptive clustering resolves initial supervision inconsistencies (e.g., caused by occlusion), we visualize clustering results before and after training.

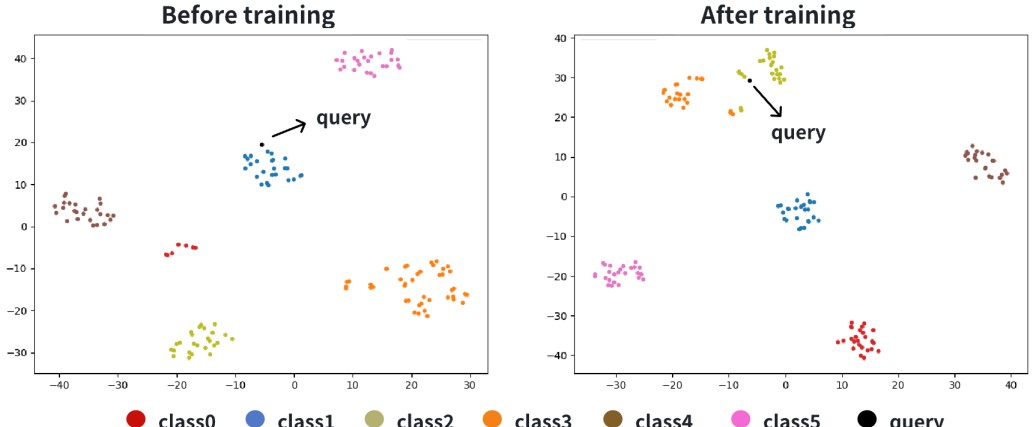

Figure 4: **The t-SNE visualization of clustering results on SF-XL-Occlusion before and after training.** The query (black dot) is reassigned from class 1 to class 2, correcting its initial misclassification.

As shown in Figure 4, we select a query (black dot) and its neighboring samples from the same geo-grid cell. Initially, the query is assigned to class 1, but after training, it transitions to class 2. These two classes correspond to adjacent view directions with overlapping semantic content.

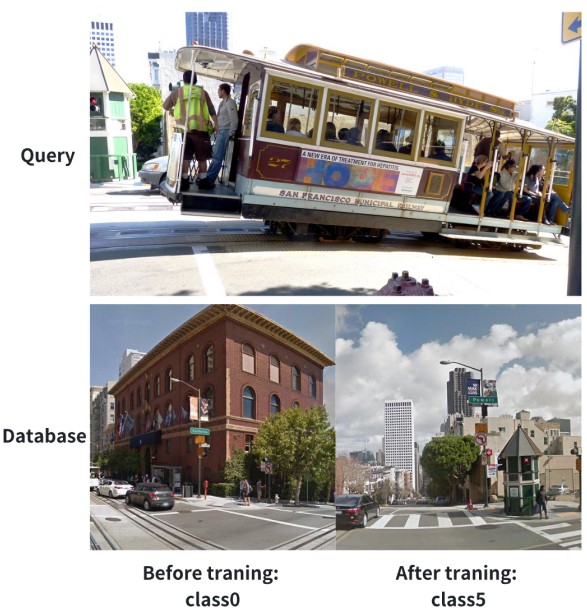

Query

Database

Before traning:
class0

After traning:
class5

Figure 5: **Close-up of the query and its nearest samples.** Visual inspection shows class 2 has stronger semantic similarity to the query.

Figure 5 shows a visual comparison of samples in class 1 and class 2, where class 2 clearly exhibits stronger visual similarity with the query. This supports our claim that adaptive clustering can realign mislabeled samples by leveraging feature similarity during training, improving robustness to occlusion and supervision noise.

This demonstrates that our adaptive clustering effectively mitigates the impact of occlusions during training by refining feature-grouping over time. It enables the model to build robust associations between partially occluded and unobstructed views from the same location.

## B.2    Feature Distance Analysis Across Classes

We further analyze how our adaptive clustering improves semantic continuity between classes by comparing feature distances of samples from adjacent categories.

To ensure fairness, we use SOTA method EigenPlaces as a proxy for selecting image pairs with adjacent labels and minimum mutual distances. As shown in Figure 6, we compare the pairwise feature distances obtained by our method and CosPlace.

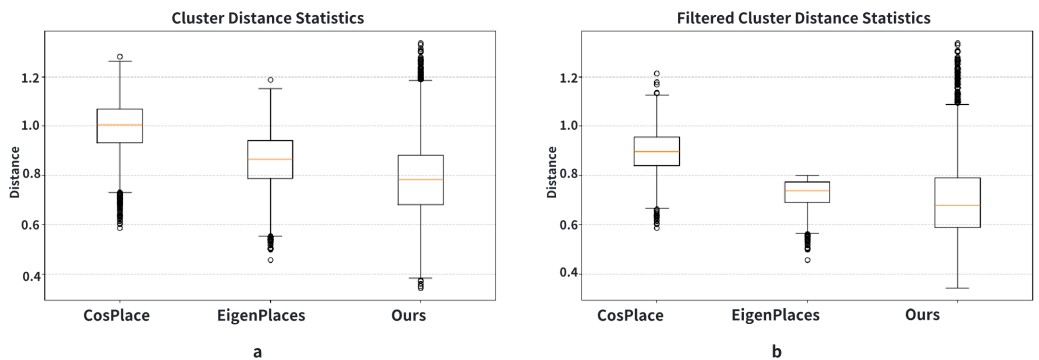

Figure 6: **Visualization results.** Feature distance comparisons between adjacent classes, using EigenPlaces as a proxy. Our method shows tighter clustering both overall (a) and for similar pairs with $d < 0.8$ (b), indicating better feature continuity across classes.

The results show that our method consistently achieves the smallest feature distances across adjacent view directions Even when we focus on sample pairs with closer feature distances (< 0.8), as shown in Figure 6 b. Our method still maintains closer feature relationships.

Our method consistently yields lower feature distances than CosPlace and EigenPlaces, indicating that our approach encourages more compact and semantically smooth transitions between neighboring classes. This explains why even occluded queries, if misclassified, can still retrieve the correct place as long as the distance remains within the retrieval threshold.

## C    More Studies

### C.1    Comparison of High-Dimensional Descriptor

| Method | Desc.dim. | SF-XL-Occlusion | | | | SF-XL-testv1 | | | |
|---|---|---|---|---|---|---|---|---|---|
| | | R@1 | R@5 | R@10 | R@20 | R@1 | R@5 | R@10 | R@20 |
| BoQ | 8192 | 49.8 | 65.8 | 70.3 | **75.9** | 82.3 | 87.9 | **91.3** | **92.9** |
| SALAD | 8192+256 | 45.4 | 62.2 | 67.1 | 74.6 | 82.4 | 86.6 | 89.2 | 90.0 |
| SALAD+CM | 8192+256 | 46.1 | 57.9 | 64.5 | 68.4 | 80.6 | 85.3 | 87.7 | 89.1 |
| **MutualVPR (Ours)** | 8192 | **51.4** | **67.5** | **72.8** | 75.8 | **82.8** | **88.9** | 90.8 | 91.4 |

Table 6: **Comparison on SF-XL-Occlusion and SF-XL-testv1.** Comparison with high-dimensional DINOv2-based methods. Our method shows limited improvement when increasing dimensions, but still achieves competitive performance. Best results are shown in **bold**, and second best are underlined.

Since our method already achieves excellent performance with a relatively low descriptor dimension (512), we further conduct a fair comparison with SOTA methods under similar dimensional settings, as shown in Table 6. The original BoQ model has a very high dimensionality of 12288, making it infeasible to evaluate on the SF-XL dataset. Therefore, we reduce its projection dimension from 384 to 256, resulting in a total descriptor dimension of 8192.

To ensure fairness, we trained BoQ, SALAD, and SALAD+CM for 30 epochs on the GSV-Cities dataset until convergence. As shown in Table 6, most methods benefit significantly from higher feature dimensions. In contrast, our method shows only a moderate improvement when increasing the dimension from low (Table 1 and Table 2) to high, possibly because our dimensionality control is achieved through a simple MLP, whereas other methods involve additional internal projection layers.

Nevertheless, our method still delivers outstanding performance, achieving the best results in both R@1 and R@5 metrics.

### C.2    Fine-tuning Strategies

To assess the effectiveness of our framework in accommodating different fine-tuning strategies, we evaluated MulConV, the method used in our work for adapting DINOv2 features, against PEFT-based approaches proposed in SelaVPR [21] and EDTformer [16]. Table 7 presents retrieval performance across multiple datasets, including Tokyo 24/7, MSLS-Val, Pitts30k, SF-XL-v1, and SF-XL-Occlusion.

| Method | Tokyo247 | MSLS-val | Pitts30k | SF-XL-v1 | SF-XL-Occlusion |
|---|---|---|---|---|---|
| SelaVPR | 90.9 / 96.1 | 86.4 / 93.6 | 89.9 / 95.8 | 75.3 / 84.3 | 40.5 / 54.1 |
| EDTformer | 87.3 / 93.7 | 85.5 / 93.8 | 89.5 / 95.7 | 76.4 / 82.7 | 38.8 / 52.0 |
| MulConV (Ours) | 92.1 / 96.5 | 89.2 / 95.1 | 90.9 / 96.4 | 80.8 / 86.4 | 47.4 / 65.8 |

Table 7: **Comparison of different fine-tuning strategies within our framework.** Metrics are R@1 / R@5.

The results demonstrate that all fine-tuning strategies achieve competitive performance, indicating that our framework effectively leverages pretrained representations. Among the evaluated methods,

MulConV consistently delivers the highest retrieval accuracy across all datasets, particularly under challenging conditions with occlusions or extreme viewpoint variations. This superior performance motivated our choice of MulConV as the fine-tuning strategy in our framework. These findings highlight both the flexibility and robustness of our approach, showing that it can accommodate various adaptation methods while benefiting most from MulConV.

### C.3 Comparison of Different Backbones

To investigate the impact of different backbone architectures on retrieval performance, we conduct experiments using VGG16, ResNet50, and DINOv2 as feature extractors under a consistent training protocol (with $K = 3$ and no differential cropping). The results are shown in Table 8.

| Backbone | Params. | Flops. | SF-XL-testv1 | |
|---|---|---|---|---|
| | | | R@1 | R@5 |
| VGG16 | 15.0m | 77.5b | 61.5 | 70.8 |
| ResNet50 | 24.6m | 21.3b | 73.9 | 81.4 |
| DINOv2 | 100.9m | 122.8b | **77.0** | **84.6** |

Table 8: **Comparisons of various backbone.** Train under different backbones when K=3 on origin dataset without multi-angle cropping. For VGG16, only the parameters of the last layer are trained. For ResNet50, only the parameters beyond the third layer are trained. For DINOv2, only the adapter module is trained.

Despite having significantly fewer FLOPs than VGG16, ResNet50 achieves much better performance (R@1 of 73.9 vs. 61.5), highlighting the advantage of deeper residual connections and stronger feature representations. DINOv2, a vision transformer pretrained with self-supervised learning, achieves the best retrieval accuracy (R@1 of 77.0 and R@5 of 84.6), even though only a small adapter is trained on top. This confirms the strong generalization and representational capacity of DINOv2 features, making it a suitable backbone for downstream place recognition tasks.

These findings support our choice of using DINOv2 in the main experiments, striking a good balance between high performance and efficient finetuning.

## D Discussion on EigenPlaces' Performance on SF-XL-testv1

Although our method generally outperforms existing approaches, it shows a slight drop on SF-XL-testv1, where EigenPlaces performs marginally better.

This arises from their methodological difference: EigenPlaces, being geometry-driven, groups views of the same focal point from different directions, thus enforcing a strong viewpoint-invariant prior that aligns well with SF-XL's panorama-cropped, multi-view structure.

In contrast, our approach clusters images purely in visual feature space. While it lacks explicit geometric constraints, it learns semantically and spatially coherent clusters that generalize more flexibly to complex scenes with occlusion, clutter, or fine-grained variations.

To verify the above observation and better understand the performance gap on SF-XL-testv1, we conducted a quantitative analysis to measure the degree of viewpoint invariance across different class construction strategies. Specifically, we sampled approximately 9k UTM grids (about 1M images) and applied k-means clustering using three methods: CosPlace, EigenPlaces, and ours. For each grid, we computed the feature distances between clusters with opposing headings (0° vs. 180°), as illustrated in Fig. 7.

The average inter-cluster distances were CosPlace: 1.1614, EigenPlaces: 1.0759, and Ours: 1.1247. These results quantitatively confirm that EigenPlaces exhibits the strongest viewpoint invariance, which explains its superior performance on benchmarks characterized by extreme viewpoint changes such as SF-XL-testv1. Conversely, CosPlace shows the largest inter-cluster distance and correspondingly lower performance, further validating that this metric meaningfully reflects a model's sensitivity

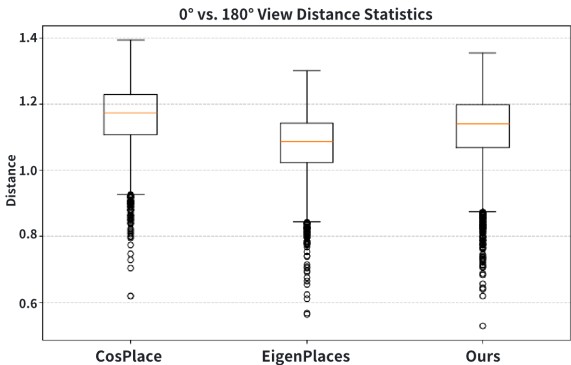

Figure 7: **Comparison of viewpoint invariance.** Box plot of inter-cluster distances (0° vs. 180°) for CosPlace, EigenPlaces, and Ours. Smaller values indicate stronger viewpoint invariance.

to viewpoint variation. Our method lies between the two, achieving a balanced trade-off—less rigid viewpoint invariance but greater robustness to occlusion, scene clutter, and appearance ambiguity.

