# OpenReview forum: "MutualVPR: A Mutual Learning Framework for Resolving Supervision Inconsistencies via Adaptive Clustering"
_NeurIPS.cc/2025/Conference — NeurIPS 2025 poster_

### Official Review · Reviewer_6zzo · 2025-06-22

**Clarity:** 3
**Significance:** 3
**Originality:** 3
**Rating:** 4
**Confidence:** 4

**Summary:**

The paper introduces mutualVPR, a method designed to address label inconsistency in classification-based Visual Place Recognition (VPR) techniques, which can degrade performance and generalization. The authors identify three main causes for label inconsistency:

1. Overlapping visually similar scenes being assigned different classes due to differing view directions.

2. Visually distinct scenes captured from similar directions being assigned the same class.

3. Views from the same reference point exhibiting varying semantic content due to occlusions.

mutualVPR aims to resolve these inconsistencies by jointly learning classification-based VPR while iteratively refining initial geo-based labels. This self-supervised refinement enforces semantic consistency of learned features through K-means clustering of visual features, which is iteratively applied as the features evolve during the learning process. The method is evaluated on general and occlusion-specific VPR benchmarks and compared to several recent baselines. The quality of the proposed clustering method is further compared to manual labelling techniques.

**Questions:**

*Questions*
1. Have you directly applied your adaptive clustering method with existing VPR models, such as CosPlace, to enable a more direct and fair comparison?

2. Have you explored applying continual learning to classification-based VPR, where initial labels are refined with your adaptive approach, rather than training from scratch?

*Suggestions*
1. Please clarify whether the VPR architecture used is an existing one or a novel contribution. If it's a new architecture, consider adding an ablation study to assess its specific contribution relative to the adaptive labeling strategy. Otherwise, it is important to evaluate it across multiple baselines.

2. Extending the discussion on the method's inferior performance on the SF-XL dataset, both in the comparative analysis and the context of the label analysis. This would help readers understand the specific scenarios where this approach may be less effective.

**Ethical Concerns:**

["NO or VERY MINOR ethics concerns only"]

**Final Justification:**

I have read the authors response as well as the comments from the other reviewers. The authors have addressed my main questions and overall reasons to accept the paper overweigh reasons to reject it - interesting problem, relatively novel approach and superior performance. I thus keep my original rating.

**Limitations:**

yes

**Quality:**

2

**Strengths And Weaknesses:**

*Strengths*

1. The paper identifies and tackles a significant limitation in the current labeling and supervision strategies for classification-based Visual Place Recognition (VPR) methods. The proposed VPR method effectively combines adaptive clustering with feature learning to mitigate label inconsistency.
2. The method demonstrates competitive performance against recent state-of-the-art classification-based and contrastive-based VPR techniques across multiple standard benchmarks. It also achieves state-of-the-art results on benchmarks specifically designed for VPR in occluded environments.
3. A comparative analysis reveals that the shows clustering-based adaptive labeling strategy outperforms manual labeling strategies used by other state-of-the-art VPR
4. The paper is well-structured and clearly written.

*Weaknesses*

1. The effectiveness of the proposed labeling method appears to be intrinsically linked to the quality of the learned features from the VPR (Visual Place Recognition) model it's coupled with. However, the paper does not compare the performance of their method across different VPR architectures, which makes it difficult to fully appreciate the specific contribution of their proposed approach.
Moreover, a key ambiguity also remains regarding whether the authors introduce a new neural network architecture in addition to their adaptive clustering method:
- If a new architecture is proposed, the absence of an ablation study comparing its performance with and without adaptive clustering limits the reader's ability to isolate the contribution of the backbone from that of the adaptive labeling.

- Conversely, if no new architecture is proposed, the specific base architecture used is not explicitly mentioned. In this scenario, it would be important to evaluate the proposed labeling method when coupled with multiple VPR baselines to demonstrate its generalizability.
2. Static K Hyperparameter: The method relies on a static K hyperparameter, which significantly influences performance (within the margin considered for outperforming other methods in the comparative analysis). The paper does not address how to estimate the optimal or expected impact of K, although the authors do acknowledge this as a limitation.
3. Inconsistent Performance Discussion: The paper provides limited discussion regarding instances where classification-based methods outperform the proposed method on the SF-XL dataset, both in the comparative analysis and when analyzing the effect of the labeling method. A deeper dive into these specific cases would strengthen the analysis.
4. The proposed method is limited to classification-based VPR methods (or at least presented as such)

---

> ### Author Rebuttal · Authors · 2025-07-31
>
> ### Response to Q1:
> Yes. CosPlace is our most direct baseline, and our work can be seen as replacing its fixed, orientation-based labeling with our adaptive clustering method. The results in Table 1 and Table 2 demonstrate the overall effectiveness of our approach. For the most direct comparison, Table 3 explicitly ablates the performance of our method against CosPlace on the same backbones (DINOv2 and ResNet50), clearly isolating and validating the significant contribution of our adaptive clustering.
>
> ### Response to Q2:
> This is a very interesting and insightful suggestion. We have not yet explored the application of our method in a continual learning setting, but we agree it is a promising direction. Using our adaptive clustering to refine labels over time as a system encounters new data could be a way to handle long-term appearance changes. We thank you for this idea and welcome further discussion on its potential.
>
> ### Response to weakness1:
> We thank you for highlighting this ambiguity. We would like to clarify: We do not propose a new architecture from scratch. Our contribution is a training framework, MutualVPR, which integrates adaptive clustering with a feature extractor in a mutual learning loop. As detailed in Section 3.1, this framework is a training methodology applied to classification-based architecture. Our novelty lies in the co-evolution of clustering and feature learning to resolve supervision inconsistency, not in the base architecture itself.
> We first demonstrate the effectiveness of our method by comparing it with current state-of-the-art approaches in Table 1, where our method consistently outperforms others across benchmarks.
>
> To isolate the contribution of our adaptive clustering, we compare with CosPlace in Table 3, which uses a fixed labeling strategy. This serves as an ablation, showing the benefit of our dynamic labeling. Additionally, the case of K=1 in Table 4, equivalent to disabling our clustering, leads to clearly worse performance, further confirming the value of our method.
>
> We acknowledge the importance of evaluating a broader set of baselines. In response, we have already included additional comparisons with both classification-based and contrastive learning methods. As reflected in Reviewer W9pq’s comments, our method still achieves state-of-the-art performance under the same feature dimensionality, demonstrating its robustness and generality. We will continue to incorporate more baseline comparisons in future revisions to further validate the effectiveness of our approach.
>
> ### Response to weakness2:
> We thank you for this point and agree that the fixed nature of K is a limitation, which we have acknowledged in the paper. We conducted an ablation study in Section 4.5.2 to analyze its impact.
>
> Generally estimate the optimal K is currently infeasible and has to be decided by ablation. However, we found that the choice of an optimal K is related to the dataset's characteristics and the backbone.
>
> On the SF-XL dataset, as regarding to the backbone, interestingly, the descriptors from ResNet50 exhibited more clearly defined cluster boundaries, while those from DINOv2 often showed overlapping clusters — in some cases, even two classes with different orientation labels appeared to belong to a single cluster. This suggests that ResNet50 tends to produce more distinct separations, whereas DINOv2 brings similar-looking images closer in the feature space.This observation corresponds well with our results in Table 4, where ResNet50 performs better with K=6, and DINOv2 performs better with K=3.
>
> While these findings provide some insights, we do not yet have a definitive criterion for selecting the optimal K. Therefore, we still regard the choice of K as a limitation of our approach.
>
> ### Response to weakness3:
> We appreciate your comments and would like to clarify that on the SF-XL-Occlusion dataset, our method achieves the best performance, which strongly supports our core motivation of preserving label consistency. Detailed discussions can be found in Section 4.4 and Appendix B.1–B.2.
>
> In contrast, EigenPlaces performs better than ours on the SF-XL-testv1 split. As noted in the main text (line 171 in the paper), we attribute this to overfitting. Upon further analysis, we find that SF-XL contains multi-view training samples. Following the EigenPlaces pipeline, images are cropped from panoramic views facing the same direction, which introduces strong viewpoint sensitivity. Since SF-XL-testv1 also consists primarily of multi-view test samples, this leads to overfitting.
>
> However, EigenPlaces’ sampling strategy does not handle occlusion well. When the viewpoint is fixed and occluded by obstacles, label inconsistency arises — a core challenge our method addresses. This is evidenced by our 10-point performance gain over EigenPlaces on SF-XL-Occlusion, highlighting its inferior generalization ability.
>
> Anyway, we agree that this reasoning should be made explicit, and we will include this analysis in the appendix for clarity.
>
> ### Response to weakness4:
> This is an insightful question. Taking contrastive learning — another representative paradigm in VPR — as an example, we can better understand the distinction between learning paradigms. In contrastive learning, samples are divided into positive and negative pairs, and the goal is to increase the distance between them. However, distances among positive samples are typically not explicitly constrained, which gives rise to a variety of hard sample mining strategies.
>
> In contrast, classification-based methods enforce clear class boundaries, naturally encouraging samples within the same class to cluster more tightly, though the separation between different classes may be less structured. As demonstrated in Section 4.4 and Appendix B.1, our method leverages this structure to pull semantically similar samples (e.g., different views of the same place) closer in the feature space, thereby improving retrieval performance.
>
> While our work is built upon the classification paradigm, we believe the core idea — dynamically aligning semantically similar instances — could be extended to enhance contrastive learning as well, for instance by informing hard negative mining with our learned clusters.
>
> This is beyond the current scope of our study, but we consider it a promising direction for future research.
>
> ### Response to sugesstions:
> Thank you for your helpful suggestions.
> For suggestion1, we have addressed it in our response to Weakness 1, and we have added new baselines. The corresponding results are included in our response to Reviewer W9pq.
>
> For suggestion2, we have responded under Weakness 3, and we will include the related discussion in the appendix.

---

> > ### Comment · Reviewer_Xqf7 · 2025-08-03
> >
> > Even though this reply was concerning to another Reviewer comments, for the sake of clarity, I would like to highlight some inaccuracies in the author rebuttal:
> >
> > > Upon further analysis, we find that SF-XL contains multi-view training samples.
> >
> > I guess the authors mean panoramic images.
> >
> > > Following the EigenPlaces pipeline, images are cropped from panoramic views facing the same direction.
> >
> > EigenPlaces builds a class by picking a focal point and cropping panoramas so they all look towards that point. So, images in a class do **not** have the same heading (which would be the case of CosPlace). Each crop has a different heading, so they all depict the focal point.
> >
> > > which introduces strong viewpoint sensitivity.
> >
> > By definition, this formulation induces a prior that makes embeddings **invariant** to viewpoint, not *sensitive*, which would mean co-variant. this is proven by the experimental results in the original paper and by PCA analysis of embedding when varying viewpoint
> >
> > >  Since SF-XL-testv1 also consists primarily of multi-view test samples, this leads to overfitting.
> >
> > The test set is made up of Flickr queries, and the database are images cropped from panoramas; hence I do not understand what 'multi-view' images means. I understand even less the concept of 'overfitting' in this scenario, in a retrieval context where only the feature extractor is used at test time. I could see as overfitting a model that performs well only in the city where it was trained on, which is not the case. Given the confusing discussion, it seems the 'overfitting' term is thrown around to justify unclear results.
> >
> > In my view, the reason why the method slightly underperforms on SF-XL testv1 is that it bases the class construction on clustering from visual-features, which makes it harder to capture heavy viewpoint variations which cause visual shift. the v1 test set is the one with most challenging viewpoint variations, and it is reasonable that EigenPlaces that purposefully targets viewpoint variations performs better in that test set. The fact that a method with a training protocol that explicitly formulates viewpoint invariance as inductive prior works better in this scenario is not 'overfitting', is an expected behavior given the priors induced by training. Honestly, I think this response from the authors lacks critical thinking and aims at dismissing a question from a reviewer with a quick reply, hoping that no one would take the time to delve in the details.
> >
> > To be clear, I appreciate the paper and its contributions, and I do not believe that slightly underperforming on one benchmark is grounds for rejection. However, I strongly encourage the authors to clarify this point and to engage in a more thorough analysis of the experimental results. A thoughtful discussion, even if it reveals limitations of the method, would strengthen the paper and demonstrate a deeper understanding of the underlying factors, rather than offering a vague explanation that risks undermining the overall rigor of the work.

---

> > ### Comment · Reviewer_6zzo · 2025-08-05
> >
> > I thank the authors for their response and for addressing my questions. My main concerns were addressed.

---

> > > ### Author Response · Authors · 2025-08-05
> > >
> > > We sincerely thank the reviewer for their thoughtful comments and for acknowledging that our responses have addressed their concerns. We appreciate the time and effort spent reviewing our work.

---

> ### Author Response · Authors · 2025-08-04
>
> We sincerely thank the reviewer for his insightful explanation regarding the strengths of EigenPlaces. We fully agree that the interpretation: linking EigenPlaces’ superior performance to its strong viewpoint-invariant inductive prior that aligns well with the SF-XL-testv1 benchmark’s structure, is accurate and much more precise than our earlier description. We appreciate the reviewer clarifying that this advantage reflects a form of task specialization rather than traditional overfitting, which helps deepen the understanding of the method’s behavior.
>
> Our main argument centers around the different inductive priors imposed by class construction strategies during training:
>
> EigenPlaces (geometry-driven) constructs classes by grouping views that observe the same focal point from different direction. This strategy imposes a strong viewpoint-invariant prior, which is particularly effective in benchmarks such as SF-XL-testv1, where the database entries differ significantly in viewing direction due to panorama cropping. The resulting performance reflects a favorable alignment between this inductive bias and the benchmark’s structural properties—what we initially (and incorrectly) referred to as “overfitting”.
>
> Our Method (appearance-driven) forms classes via clustering in visual feature space, without relying on geometric information. While this does not explicitly enforce viewpoint invariance, the clusters are often semantically or visually coherent. Moreover, neighboring clusters typically correspond to spatially adjacent views, offering flexibility and resilience in complex scenarios involving occlusion, scene clutter, or subtle semantic variation.
>
> To quantify this distinction, we sampled approximately 6k UTM grids (~0.7M images). For each grid, we performed k-means clustering based on both EigenPlaces and our method, and measured the feature distances between clusters of opposing headings (0° vs. 180°). On average, EigenPlaces exhibited the smallest distances (1.022), followed closely by ours (1.097), confirming its stronger viewpoint invariance, meaning that extreme viewpoints have closer feature distances and are thus more likely to be correctly recalled.
> This experiment confirms that EigenPlaces effectively encodes viewpoint invariance, explaining its superior performance on benchmarks with extreme viewpoint changes like SF-XL-testv1.
>
> However, it is important to emphasize that our method is not devoid of viewpoint invariance; rather, it adopts a different inductive bias and is not specifically specialized for handling such extreme viewpoint variations.
>
> Yet, as shown in Table 2 and Appendix B.2, our method demonstrates stronger performance in visually complex environments, such as those with heavy occlusion or clutter, thanks to its finer-grained and adaptive grouping of nearby viewpoints. This advantage is reflected in smaller feature distances under moderate viewpoint shifts, indicating better local viewpoint consistency and robustness to partial occlusion.
>
> We believe this highlights a meaningful trade-off: EigenPlaces leverages a structured, geometry-based prior that enforces strong global invariance, whereas our method promotes adaptability through appearance-driven learning, which may be better suited for unstructured, occlusion-heavy, or cluttered scenes. Importantly, our method also effectively handles viewpoint variations, though with a different inductive bias than EigenPlaces.
>
> Due to the submission policy, we are unable to provide visualization results at this stage. However, we promise to discuss this aspect and include the corresponding experiments in our final version.
>
> We sincerely appreciate your clarification and the use of more precise terminology to help express our point more accurately than we initially did. We're also grateful that you found value in our work. If there are any remaining concerns, we'd be happy to further discuss them. We genuinely hope you enjoyed our submission and would kindly consider a higher score if you feel our clarifications have addressed your concerns.

---

### Official Review · Reviewer_W9pq · 2025-06-24

**Clarity:** 3
**Significance:** 3
**Originality:** 3
**Rating:** 4
**Confidence:** 4

**Summary:**

The authors identify and visualize an interesting problem in existing classification-based Visual Place Recognition (VPR) methods: inconsistent supervision caused by significant viewpoint variations and occlusions. This paper proposes a mutual learning framework that integrates unsupervised K-means clustering with feature descriptor training. Rather than relying on fixed or heuristically defined place labels, the proposed method defines and refines place labels through adaptive clustering in the embedding space. A MulConv adapter is incorporated into the frozen DINOv2 backbone to extract multi-scale features. In the mutual learning loop, clusters are updated by K-means while the extractor achieves consistent supervision by the posterior classification. Comparative experiments of various methods on popular benchmarks demonstrate excellent performance, especially on the dataset with severe occlusions.

**Questions:**

Please refer to weaknesses.

**Ethical Concerns:**

["NO or VERY MINOR ethics concerns only"]

**Final Justification:**

The author's detailed explanation and additional experiment results are appreciated and address most of my concerns. I will maintain my original rating for the manuscript.

**Limitations:**

Yes, limitations have been included as a part of the paper.

**Paper Formatting Concerns:**

I haven't seen any formatting issues in the manuscript.

**Quality:**

3

**Strengths And Weaknesses:**

Strengths:
1. The overall writing is satisfied, the authors clearly illustrate the limitations of existing classification-based visual place recognition baselines, and then propose their framework to address these issues with convincing qualitative and quantitative results.
2. This paper focuses on an interesting yet important issue of existing classification-based VPR methods, *i.e.*, the supervision inconsistency due to their fixed or heuristically defined place labels, and proposes a novel mutual learning framework that dynamically updates both feature representations and place label assignments.
3. The experiments look good, demonstrating the superiority of the proposed method by comparison with both contrastive learning-based and classification-based baselines.

Weaknesses:
I haven't seen any major weakness, some minor issues are listed as follow:
1. Compared to some recent baselines in this paper such as CricaVPR and BoQ, the number of baselines for comparison with the proposed method is less, hence it is suggested to include more baselines to present a comprehensive comparison, such as GeM [1], SFRS [2], Conv-AP [3], GCL [4].
2. The BoQ baseline used 4096 and 16384 dimensional descriptors for comparison in their paper, so it is strange why this paper re-trained BoQ with only 512 dimensional descriptors for comparison while other baselines (MixVPR, CricaVPR) use 4096 dimensional descriptors. And it is suggested to evaluate the proposed MutualVPR with diversified dimensional descriptors to demonstrate its effectiveness.
3. The authors omit the reference of GeM [1] in this paper, which is used as a part in the proposed framework.
4. The introduction of the modules in the proposed framework is too brief, such as the GeM pooling and the LMCL classifier, more description and explanation of why they are suitable for this framework are beneficial.
5. The colors of different classes in the left and the right panels of Figure 2 are not consistent.
6. The authors omit the information of training hyper-parameters, such as the learning rate, the learning scheduler, and the optimizer.

[1] Filip Radenovi´c,  et al. Fine-tuning CNN image retrieval with no human annotation. IEEE Transactions on Pattern Analysis and Machine Intelligence (T-PAMI), 2018.

[2] Yixiao Ge, et al. Self-supervising fine-grained region similarities for large-scale image localization. European conference
on computer vision (ECCV), 2020.

[3] Amar Ali-bey, et al. GSV-CITIES: Toward appropriate supervised visual place recognition. Neurocomputing, 2022.

[4] Mar´ıa Leyva-Vallina, et al. Data-efficient large scale place recognition with graded similarity supervision. IEEE/CVF Conference on Computer Vision and Pattern Recognition (CVPR), 2023.

---

> ### Author Rebuttal · Authors · 2025-07-31
>
> ### Response to weakness1:
> Thank you for this suggestion to broaden our experimental comparison. We have conducted additional experiments, and the results for GeM, Conv-AP, SALAD, and AnyLoc are presented below. Following Reviewer uVwD’s recommendation, we also include additional results on the Pitts250k-test dataset. Furthermore, in response to weakness2, we include results for CricaVPR (512-dimensional) under the same training and evaluation settings as our main method.
>
> Our method continues to achieve state-of-the-art performance at an equivalent descriptor dimension and notably demonstrates a substantial advantage on the challenging SF-XL-Occlusion dataset.
>
> For the baseline setups:
> - GeM: We trained a ResNet50 backbone on the GSV-Cities dataset for 40 epochs.
> - SALAD (512+32): We trained a DINOv2 backbone for 40 epochs on GSV-Cities to ensure consistency with our BoQ (512) training.
> - CricaVPR: We applied PCA to reduce the descriptor dimensionality to 512 for a fair comparison.
> - AnyLoc: The original implementation uses ViT-G/14 with VLAD or GeM. However, ViT-G/14 incurs slow inference on the SF-XL dataset. To maintain practical and comparable settings, we used ViT-B/14 with GeM for evaluation, which provides a balanced trade-off between performance and efficiency.
>
> The results are as follows:
>
> | Method              | Dim        | Trainset     | Tokyo247      | MSLS-val       | Pitts30k-test  | Pitts250k-test | SF-XL-v1    | SF-XL-Occlusion |
> |---------------------|------------|--------------|----------------|----------------|----------------|----------------|----------------|------------------|
> | NetVLAD             | 131072     | Gsv-cities   | 69.5 / 82.5    | 82.8 / 90.3    | 87.0 / 94.3    | 89.1 / 94.8    | -              | -                |
> | GeM                 | 2048       | Gsv-cities      | 60.3 / 73.7    | 72.5 / 82.7    | 84.5 / 92.8    | 85.1 / 93.4    | 25.6 / 35.5    | 11.8 / 15.8      |
> | AnyLoc   | 768      | -            | 71.7 / 87.6    | 32.6 / 41.6    | 77.7 / 88.9    | 79.3 / 89.5    | 33.3 / 45.2    | 6.6 / 14.5       |
> | ConvAP              | 2048       | Gsv-cities   | 74.6 / 83.2    | 81.5 / 87.5    | 89.7 / 95.2    | 91.2 / 96.4    | 41.1 / 53.0    | 23.7 / 26.3      |
> | MixVPR              | 4096       | Gsv-cities   | 87.0 / 93.3    | 87.1 / 91.4    | 91.6 / 95.5    | 94.3 / 98.1    | 69.2 / 77.4    | 30.3 / 35.5      |
> | CricaVPR*           | 4096       | Gsv-cities   | 93.0 / 97.5    | 90.0 / 95.4    | 94.9 / 97.3    | -              | -              | -                |
> | CricaVPR1           | 4096       | Gsv-cities   | 89.5 / 94.6    | 88.5 / 95.1    | 91.6 / 95.7    | 94.3 / 98.6    | 72.8 / 80.1    | 40.8 / 51.3      |
> | CricaVPR1           | 512(PCA)        | Gsv-cities   | 87.4 / 92.9    | 87.1 / 92.6    | 90.4 / 94.9    | 92.5 / 97.1    | 68.4 / 77.1    | 36.9 / 48.1      |
> | BoQ†                | 512        | Gsv-cities   | 91.9 / 95.5    | 88.4 / 93.9    | 93.1 / 96.1    | 93.8 / 97.5    | 79.6 / 85.9    | 38.2 / 50.0      |
> | SALAD               | 512 + 32   | Gsv-cities   | 92.3 / 95.1    | 88.5 / 94.2    | 90.6 / 95.1    | 92.1 / 97.0    | 70.2 / 77.7    | 31.6 / 42.1      |
> | EigenPlaces         | 512        | SF-XL        | 84.8 / 94.0    | 88.1 / 92.9    | 92.3 / 96.1    | 93.5 / 97.5    | 83.8 / 89.6    | 36.8 / 51.8      |
> | CosPlace            | 512        | SF-XL        | 76.5 / 89.2    | 84.4 / 90.2    | 89.6 / 94.9    | 90.4 / 96.6    | 64.8 / 73.1    | 32.9 / 43.4      |
> | MutualVPR (Ours)    | 512        | SF-XL        | 92.1 / 96.5    | 89.2 / 95.1    | 90.9 / 96.4    | 92.6 / 97.9    | 80.8 / 86.4    | 47.4 / 65.8      |
>
> All methods will be included in the final manuscript, along with a detailed description of the experimental setup.
>
> Since GCL and SFRS are difficult to scale for training on the GSV-Cities dataset, we do not include them in the main experiments to ensure fairness. However, in response to the reviewer’s suggestion, we present their testing results and experimental settings here for reference.
>
> For GCL(ResNet50+GeM), we directly used the official pretrained weights and evaluated under our testing protocol.
> SFRS has a high feature dimensionality (32768), which makes it difficult to test on the SF-XL dataset. Therefore, we used the official pretrained weights and applied PCA to reduce the dimension to 4096. However, this dimensionality reduction led to a significant performance gap compared to the results reported in the original paper.
>
> The results are as follows:
>
> | Method | Train-set | Dim       | Tokyo247   | MSLS-val   | Pitts30k-test | Pitts250k-test | SF-XL-v1   | SF-XL-Occlusion |
> |--------|-----------|-----------|------------|------------|---------------|----------------|------------|-----------------|
> | SFRS   | MSLS      | 4096 (PCA)| 36.2 / 50.8| 58.2 / 66.6| 76.0 / 86.8   | 78.2 / 88.2    | 14.0 / 21.9| 5.3 / 11.8      |
> | GCL    | Pitts250K | 2048      | 42.3 / 60.8| 64.0 / 76.6| 71.3 / 86.7   | 67.4 / 83.7    | 4.1 / 8.3  | 2.6 / 5.3       |
>
> ### Response to weakness2:
> Thank you for highlighting this important point.
>
> Our primary baselines are CosPlace and EigenPlaces, both of which typically use 512-dimensional descriptors. To ensure consistency and fairness, we adopted the same dimensionality across our main comparisons.
>
> For BoQ, which also uses DINOv2 as the backbone, we chose 512 dimensions to allow for a fair and consistent comparison under the same training and evaluation settings. Although CricaVPR also uses DINOv2, its performance is known to be sensitive to batch size. Since our evaluation uses a batch size of 1, we followed the official CricaVPR setup and retained the 4096-dimensional descriptors to match their published settings.
>
> In response to your suggestion, we have now included CricaVPR (512) in our evaluation, and the results are provided in our response to weakness1.
>
> For methods such as MixVPR, which use ResNet50, we retained their original feature dimensionality to preserve the integrity of their published configurations.
>
> Additionally, we are currently training a version of our method using 4096-dimensional descriptors to explore performance at higher dimensions. However, due to time constraints during the rebuttal period, training is still in progress and results are not yet available. We will include these results and discussions in the final version of the manuscript.
>
> ### Response to weakness3:
> The omission of the GeM reference was our mistake. We have added GeM as a baseline and will include the proper citation.
>
> ### Response to weakness4:
> Thank you for this suggestion. Our use of LMCL classifier follows the precedent set by CosPlace, as we mentioned in the final paragraph of Section 3.2.2. Similarly, we employed GeM pooling as it is a standard and widely-adopted aggregation method in the VPR field.
>
> We intentionally kept the descriptions of these modules concise, as our primary focus is on our mutual learning framework rather than  established components. However, we agree with your that providing more detail would improve the paper's accessibility for readers who may not be deeply familiar with this specific domain.
>
> Therefore, we will accept your valuable suggestion and add a more detailed description of both the LMCL classifier and GeM pooling to the appendix in the final manuscript.
>
> ### Response to weakness5:
> Thank you for catching the color inconsistency in Figure 2; we will correct it in the revision.
>
> ### Response to weakness6:
> We will add all relevant training parameters to the "Implementation Details". Thank you for your suggestion.

---

> ### Comment · Reviewer_W9pq · 2025-08-01
>
> The author's detailed explanation and additional experiment results are appreciated and address most of my concerns. I have understood the inconvenience to train your model with 4096-dim descriptors due to time constraints of the rebuttal period, so I would expect related experimental results in the future revised version. I will maintain my original rating for this manuscript.

---

### Official Review · Reviewer_Xqf7 · 2025-06-25

**Clarity:** 2
**Significance:** 3
**Originality:** 3
**Rating:** 5
**Confidence:** 5

**Summary:**

The paper proposes a novel training protocol for VPR. Existing works that adopt a classification framework rely on UTM coordinates for assigning samples to classes, which is prone to perceptual aliases. The paper introduces an unsupervised strategy based on feature clustering to adaptively divide into classes samples within a given geographical area. The architecture entails a frozen Dinov2 backbone with parameter-efficient tuning

**Questions:**

- in the experimental section it is not clear whether the proposed framework is compared with EigenPlaces and CosPlace with the original ResNet50, or with Dinov2 as the proposed method. the backbone should be the same for fair comparison
- an ablation to study the effect of finetuning DINO vs PEFT should be included
- at training time, why is a frozen feature extractor employed for clustering? this would double the amount of forward passes required. it is also common in contrastive netvlad-style training to use the same feature encoder updated as training procedes. is there any reasoning or experiment behind this choice?

**Ethical Concerns:**

["NO or VERY MINOR ethics concerns only"]

**Final Justification:**

The main concerns I had with the paper were writing clarity (the training set construction was not self-contained, in my opinion), as well as clarity in the adopted baselines for experimentation (backbone of CosPlace, EigenPlaces, which are highly relevant papers), and missing citations.
The main strengths of the paper are the proposal of a clustering-based training protocol that foregoes the need for heading labels, and the strong results especially on Night and Occlusion domain.

The rebuttal saw a thorough discussion, which mainly revolved around (i) providing a better discussion to explain experimental data, especially wrt EigenPlaces which is the most related work, (ii) experimental setup for SALAD, CM, CosPlace baselines and (iii) fine tuning strategy vs adaptation for Dinov2.
The initial arguments from the authors raised some concerns, however after discussion the authors provided a clear and thorough analysis of the problem, which solved the main issues for me.

I highly suggest to the authors to incorporate the outcome of the discussion on the above topics in the final version, as well as the missing citations.

**Limitations:**

yes

**Paper Formatting Concerns:**

-

**Quality:**

3

**Strengths And Weaknesses:**

Strenghts
- the proposed training protocol removes the need for heading labels, which can be hard to obtain in many practical scenarios
- Compared to recent state of the art (which are not cited btw, see weaknesses), which finetune DINO and use large output descriptors, the proposed method keeps DINO frozen, trains only lightweight adapters, and keeps dimensionality compact.
- the results on SF-XL occlusion, and the comparison with manual heading supervision are interesting and show the value of the proposed algorithm
- the proposed method improves upon Eigenplaces with a solid framework, removing the need for a lateral and frontal loss.

Weaknesses:
- literature study: the literature review is missing several relevant works. [1] is relevant for classification based frameworks, it also avoids using heading labels. [2] is relevant for view-dependent sample mining. [3] is the missing citation for SF-XL occlusion dataset. [5, 6] represent the state of the art in VPR and are not compared. Although they use much larger descriptors should be considered. [4] uses parameter-efficient tuning on top of DINO
- writing clarity: throughout the paper a more consistent notation could be used. For example, when 'classes X and. Y' are mentioned by number, it is hard for a non-expert in the task to grasp the referred concept. Also, the related works section feels misplaced in the end of the manuscript. Section 2, "problem analysis', could also benefits from more formal writing. I could understand the content given that I worked extensively on the task, but to make the paper more accessible some things should be made clearer, e.g. 'manual labelling' is misleading, it should be made clear that it means using heading labels with fixed splits. Same goes for the notation 'VIew X' and 'Class X'

[1] Trivigno, Gabriele, et al. "Divide&classify: Fine-grained classification for city-wide visual geo-localization." Proceedings of the IEEE/CVF International Conference on Computer Vision. 2023.

[2] Leyva-Vallina, María, Nicola Strisciuglio, and Nicolai Petkov. "Data-efficient large scale place recognition with graded similarity supervision." Proceedings of the IEEE/CVF Conference on Computer Vision and Pattern Recognition. 2023.

[3] Barbarani, Giovanni, et al. "Are local features all you need for cross-domain visual place recognition?." Proceedings of the IEEE/CVF Conference on Computer Vision and Pattern Recognition. 2023.

[4] Lu, Feng, et al. "Towards seamless adaptation of pre-trained models for visual place recognition." arXiv preprint arXiv:2402.14505 (2024).

[5] Izquierdo, Sergio, and Javier Civera. "Optimal transport aggregation for visual place recognition." Proceedings of the ieee/cvf conference on computer vision and pattern recognition. 2024.

[6] Izquierdo, Sergio, and Javier Civera. "Close, But Not There: Boosting Geographic Distance Sensitivity in Visual Place Recognition." European Conference on Computer Vision. Cham: Springer Nature Switzerland, 2024.

---

> ### Author Rebuttal · Authors · 2025-07-31
>
> ### Response to Q1:
> Thank you for raising this important point.
>
> In Table 1, we followed the original implementations of CosPlace and EigenPlaces, both of which use ResNet50 as the backbone. To ensure a fair comparison, we additionally performed ablation studies in Table 3, where MutualVPR and CosPlace are compared using both ResNet50 and DINOv2 backbones under consistent data settings.
>
> We acknowledge that our comparison does not include a version of EigenPlaces with a DINOv2 backbone. While we attempted such experiments, the results were notably suboptimal (e.g., Tokyo247: R@1 = 81.4, R@5 = 90.2; SF-XL-testv1: R@1 = 75.3, R@5 = 83.1). We suspect this may be due to the sampling strategy and fine-tuning procedures employed in EigenPlaces, which may not be compatible with the representational characteristics of DINOv2. Consequently, these results may not reflect the backbone’s true potential. To avoid possible misinterpretation, we opted not to include these results in the main paper. We will clarify this rationale in the final version for transparency.
>
> ### Response to Q2:
> Thank you for this valuable suggestion.
>
> As fine-tuning methods are not the core focus of our work, we did not include a detailed comparison in the main paper. However, we had previously conducted experiments comparing MulConV with the PEFT method proposed in [1], and we ultimately chose MulConV due to its superior performance in our setting.
>
> In response to your comment, we have now conducted additional experiments using the PEFT method from [2].  The following are our experimental results:
> | Method       | Tokyo247     | MSLS-val     | Pitts30k-test | SF-XL-v1     | SF-XL-Occlusion |
> |--------------|--------------|--------------|----------------|--------------|------------------|
> | SelaVPR[1]   | 90.9 / 96.1  | 86.4 / 93.6  | 89.9 / 95.8    | 75.3 / 84.3  | 40.5 / 54.1      |
> | EDTformer[2] | 87.3 / 93.7  | 85.5 / 93.8  | 89.5 / 95.7    | 76.4 / 82.7  | 38.8 / 52.0      |
> | Ours     | 92.1 / 96.5 | 89.2 / 95.1 | 90.9 / 96.4 | 80.8 / 86.4 | 47.4 / 65.8 |
>
> The results show that MulConV remains better suited to our framework, though other PEFT approaches also perform strongly. This highlights both the effectiveness and flexibility of our proposed framework in accommodating different fine-tuning strategies.
>
> We will include the results and discussion of all these fine-tuning methods in the final version of the manuscript. Thank you again for your insightful feedback.
>
> [1]Lu F, Zhang L, Lan X, et al. Towards seamless adaptation of pre-trained models for visual place recognition[J]. arXiv preprint arXiv:2402.14505, 2024.
>
> [2]Jin T, Lu F, Hu S, et al. EDTformer: An Efficient Decoder Transformer for Visual Place Recognition[J]. IEEE Transactions on Circuits and Systems for Video Technology, 2025.
>
> ### Response to Q3:
> For Q3, you are correct — our method does involve two forward passes.
> To clarify, the UTM-based geographic grid is predefined. At the start of each epoch, we sample a subset of UTM grid cells, each of which contains images captured from approximately the same location. For each selected grid cell, we group its images and perform clustering within the group to assign pseudo-labels. We extract features using the current model without gradients (i.e., the feature extractor is frozen during clustering), then assign labels based on clustering results. Once labels are updated, standard supervised training proceeds as usual.
>
> Here is the corresponding pseudocode:
>
>     for epoch in epochs:
>         sampled_grids = sample_grid_cells(dataset)
>         for grid in sampled_grids:
>             images = get_images_from_grid(grid)
>             with torch.no_grad():
>                 features = model(images)
>             labels = cluster(features)
>             update_labels(images, labels)
>         for iteration in iterations:
>             train_step()  # standard training with updated labels
>
> The motivation is that at the early stage, clustering based on initial features may assign visually similar images to different classes or visually dissimilar images to same classes, leading to label inconsistency. By updating the pseudo labels progressively during training, we ensure that the grid-based labels and the learned feature space evolve together, resulting in mutual reinforcement.
>
> We have conducted supporting experiments to validate our method. Specifically, we selected queries within the same geographic grid and visualized their assigned labels before and after training. Through multiple trials, we observed that some queries were assigned to different classes after training, indicating that label updates did occur. We discussed the benefits of this strategy in Section 4.4, and the corresponding visualizations can be found in Appendix B.1.
>
> ### Response to weakness1:
> Thank you for highlighting these important references! We will incorporate them into our Related Works section. Specifically, we will add [5] to our comparison results. Regarding [6], while it presents an interesting sample mining strategy, it is tailored for triplet construction in contrastive learning by mining positive samples. This approach is not directly applicable to our classification-based framework, which does not rely on explicit positive/negative sampling.
>
> We will also include [1–4] in the revised manuscript, including [3] as the citation for the SF-XL Occlusion dataset and [4] for its relevance in parameter-efficient fine-tuning.
>
> ### Response to weakness2:
> We appreciate your constructive feedback on the manuscript’s clarity. We agree that certain parts—particularly the “Problem Analysis” section—can be improved for readability.
>
> Since our clustering visualizations are based on class-level groupings, we believe it is appropriate to retain the term “Class.” However, to reduce confusion, we will clearly define “View X” and “Class Y” prior to our discussion on label inconsistency. Specifically: “View X” refers to the camera viewing direction derived from CosPlace’s heading-based orientation labels (i.e., fixed labels), indicated by Roman numerals. “Class Y” refers to the groups obtained through our clustering process.
>
> We will revise this explanation in both the main text and Figure 2 to improve clarity for readers less familiar with the task. We also welcome further feedback on how to improve this section.
>
> Additionally, we will adopt your suggestion to clarify that “manual labeling” refers specifically to using heading-based labels with fixed splits. If you deem it necessary, we are happy to relocate the Related Works section to follow the Introduction for better flow.

---

> > ### Comment · Reviewer_Xqf7 · 2025-08-03
> >
> > I thank the author for the rebuttal. Honestly, I would have appreciated comparisons with SALAD and CliqueMining.
> > I also have some doubts about results of CosPlace and EigenPlaces which seem to be different from that reported in the original papers. When using Dinov2+CosPlace, did you train the backbone?
> > Lastly, I do not find sufficient the motivation that 'fine-tuning' is not the motivation of our paper. I think when using/proposing an adapter a comparison to fine tuning of the last layers is necessary

---

> ### Author Response · Authors · 2025-08-04
>
> ### On Comparisons with SALAD and CliqueMining:
>
> We thank the reviewer for emphasizing the importance of comparisons with SALAD and CliqueMining (CM). We have included SALAD results in our response to Reviewer W9pq’s Q1.
>
> Regarding CM, we would like to clarify that it is a sample mining strategy specifically designed for contrastive learning. In the original CM paper, CM was used to mine training data from MSLS, and SALAD was subsequently trained on this mined dataset (MSLS + GSV-Cities). While we have not previously evaluated SALAD+CM ourselves, we agree this is a valuable comparison. However, it is worth noting that other contrastive learning baselines are evaluated on standard datasets without additional sample mining. Applying CM solely to SALAD could introduce inconsistencies in comparative evaluation.
>
> We acknowledge SALAD+CM as a strong SOTA method. To provide a fair comparison, we retrained SALAD+CM with 512+32D embeddings using the original SALAD configuration. The results are as follows:
>
> | Method           | Train-set      | Dim      | Tokyo247   | MSLS-val    | Pitts30k-test | Pitts250k-test | SF-XL-v1    | SF-XL-Occlusion |
> |------------------|----------------|----------|------------|-------------|----------------|-----------------|-------------|------------------|
> | SALAD            | GSV-Cities     | 512+32   | 92.3 / 95.1 | 88.5 / 94.2  | 90.6 / 95.1     | 92.1 / 97.0      | 70.2 / 77.7 | 31.6 / 42.1       |
> | SALAD+CM         | MSLS+GSV-Cities| 512+32   | 92.8 / 96.2 | 90.4 / 96.2  | 90.9 / 95.9     | 93.2 / 97.8      | 78.4 / 85.4 | 40.8 / 53.7       |
> | MutualVPR (Ours) | SF-XL          | 512      | 92.1 / 96.5 | 89.2 / 95.1  | 90.9 / 96.4     | 92.6 / 97.9      | 80.8 / 86.4 | 47.4 / 65.8       |
>
> While SALAD+CM achieves higher R@1 on certain datasets, our method (MutualVPR) consistently achieves superior R@5 performance across most benchmarks. Notably, on the challenging SF-XL-Occlusion dataset, our approach exhibits a clear advantage in both R@1 and R@5, demonstrating robustness in visually complex and occluded scenarios.
>
> ### Training CosPlace with DINOv2:
> We trained CosPlace with the DINOv2 backbone and MulConv adapter—consistent with our method—to ensure a fair comparison and to isolate the effects of our training pipeline. Details are provided in Section 4.5.1.
>
> ### On Differences from Reported CosPlace and EigenPlaces Results:
> We re-trained CosPlace and EigenPlaces ourselves following their original configurations. Minor discrepancies from previously published results may stem from differences in image preprocessing (e.g., resolution, normalization). However, to ensure fair comparisons, all baselines in our experiments were evaluated under identical preprocessing pipelines. If further clarification is needed, we are happy to provide additional details.
>
> ### Fine-tuning:
> We appreciate the reviewer’s recognition of our fine-tuning approach. We agree that ablation studies are important and have discussed various strategies in Reviewer Xqf7’s Q2.
>
> We experimented with fine-tuning the last DINOv2 layers using GeM pooling, testing both the last two and last four transformer blocks. However, we observed minimal improvement in training loss and significant degradation in validation performance—particularly when fine-tuning the last four layers, which resulted in faster and more severe performance drops. We attribute this to catastrophic forgetting: modifying deeper backbone layers tends to overwrite pre-trained representations, leading to instability and poor generalization.
>
> In contrast, fine-tuning the last layers of ResNet50—used in both our method and CosPlace—yielded strong performance. We hypothesize this is due to ResNet50 being pretrained on supervised tasks, which produce more task-aligned and stable features. DINOv2, being self-supervised, learns more general-purpose representations that are more sensitive to parameter updates, making naive fine-tuning less effective. Similar instability was observed when fine-tuning the last few layers of DINOv2 in CosPlace and EigenPlaces.
>
> Thus, we advocate for adapter-based fine-tuning in classification-based VPR methods, which better preserves stability and generalization performance.
>
> If you have any further questions or concerns, we would be happy to discuss them with you.

---

> > ### Comment · Reviewer_Xqf7 · 2025-08-05
> >
> > I appreciate the author's rebuttal. I still would have appreciated a comparison with salad original descriptors and I do not agree that CLiqueMining should be applied to each method in order to be comparable. Each method has its value proposition and should be compared to the others in terms of generalization across datasets.
> > I appreciate the effort to train salad at 512-D, but I think the takeaway is that SALAD and SALAD-CM still perform better (as they should be compared at original dimensionality), although the proposed method has low dimensionality which is good.
> >
> > I find it interesting that finetuning DINO does not converge, as it is the strategy adopted in SALAD & co. It would be interesting to further analyze if this is due to the aggregation of choice (NetVLAD-style in SALAD vs GeM pooling)
> >
> > Overall I find the paper contribution valuable, and I plan on keeping my original score.

---

> > > ### Author Response · Authors · 2025-08-05
> > >
> > > We sincerely thank the reviewer for the valuable comments. We acknowledge that our current descriptor size (512) is much smaller than SALAD’s original (8448). We plan to train our method with a similar dimension for a fairer comparison.
> > >
> > > We would also like to share an additional observation concerning fine-tuning the last few layers of DINOv2. When we doubled the batch size, fine-tuning the last two transformer blocks—although initially unstable with a drop in validation performance—began to improve after several epochs, with training loss decreasing and validation accuracy gradually rising.
> > >
> > > We think it supports our hypothesis that smaller batch sizes may lead to unstable gradient updates, inadvertently disturbing the pretrained DINOv2 parameters. Larger batch sizes increase the amount of data processed per update, providing more stable gradient signals that allow the model to adapt progressively without catastrophic forgetting. Nonetheless, fine-tuning the last four blocks still failed to converge, likely due to excessive disturbance of the pretrained parameters.
> > >
> > > Interestingly, we did not observe such instability in SALAD, which is trained within a contrastive learning framework. Given that our method is classification-based, we suspect these differences stem from the fundamentally distinct training paradigms. In particular, the classification paradigm may require sufficiently large data batches per update to achieve stable fine-tuning. Although this investigation goes beyond the scope of the current work, we intend to explore applying SALAD in classification settings (including ours, CosPlace, EigenPlaces) to further understand this phenomenon.
> > >
> > > We are truly grateful for the reviewer’s recognition of our work and thoughtful feedback, which have been invaluable in improving our study.

---

### Official Review · Reviewer_uVwD · 2025-06-29

**Clarity:** 4
**Significance:** 3
**Originality:** 3
**Rating:** 4
**Confidence:** 4

**Summary:**

MutualVPR is proposed to deal with inconsistent supervision signals used in current VPR methods. Various view directions and occlusions lead to unstable visual features and inconsistency. This approach is proposed to adaptively refine the image representations according to geometric grids clustered automatically, rather than manual labeling. Accurate and robust SOTA experimental results are shown.

**Questions:**

1. How does the proposed labelling strategy facilitate the VPR task using the recall@k metric w.r.t. UTM coordinates? Relations between the geo-grid labelling and the commonly used test labels.
2. In Table 1, the descriptor dimensionality of NetVLAD is incorrect. Also, I assume the NetVLAD descriptors can be precomputed offline and compared on SF-XL-testv1. Pitts250k-test is recommended to be included in the tests.
3. Question about the choice of the number of clusters. For all experiments in the paper, do you always validate them separately for all datasets? How sensitive is the number of clusters for the training? and how different the optimal K is among different datasets.

**Ethical Concerns:**

["NO or VERY MINOR ethics concerns only"]

**Final Justification:**

It will be good to include the answers in the revised version, eg, how the NetVLAD is retrained. Most of my concerns are addressed, while limitations remain in the proposed method. Thus, I will keep my score.

**Limitations:**

Yes

**Paper Formatting Concerns:**

None.

**Quality:**

4

**Strengths And Weaknesses:**

Strengths:
1. A novel labelling method to align the supervision with their geo locations. It simultaneously learns the clustering and the visual representations. Intra-grid classification shows a huge impact.
2. This method shows good improvement in the VPR task on the testing datasets, especially the one with occlusions.
3. The paper is well-written and easy to read.

Weakness:
1. Missing comparison with other DINOv2-based VPR methods, such as AnyLoc and SALAD.
2. worse than EigenPlaces on SF-XL test v1.
3. As mentioned in the paper, the number of K is still an open question.

---

> ### Author Rebuttal · Authors · 2025-07-31
>
> ### Response to Q1:
> We thank the reviewer for this insightful question, which allows us to clarify an important aspect of our classification-based approach to VPR. We believe there may be a slight misunderstanding regarding the role of the classifier and the use of labels during inference.
>
> As noted in line 121 of the paper, the classifier is used only during training to guide the network in learning discriminative features. At inference time, we do not use the classifier or any form of  "test labels". Instead, retrieval is performed directly using the learned feature descriptors via distance.
>
> Regarding how our labeling strategy impacts recall@k, our dynamic geo-grid labeling ensures that images of the same place, even if captured from diverse and inconsistent viewpoints, may be assigned to different clusters during training. However, since these clusters are geometrically close, the learned descriptors for such images remain nearby in the embedding space. As a result, during retrieval, these images are still within the distance threshold and are correctly recalled.
>
> This design enhances recall@k by increasing descriptor robustness to viewpoint variation. We have discussed this mechanism in line 203 of the paper and further supported it through visualizations and quantitative results in Appendix B.2.
>
> ### Response to Q2:
> Thank you for pointing this out. We clarify that the 131,072-dimensional NetVLAD descriptor reported in Table 1 is correct in our context. Specifically, we re-trained NetVLAD on the GSV-Cities dataset using a ResNet50 backbone (which outputs 2048-dimensional features) and the standard configuration of 64 cluster centers, resulting in a descriptor of size 2048 × 64 = 131,072.
> The original NetVLAD setup uses a VGG16 backbone with 512-dimensional features and 64 clusters, leading to a descriptor of 512 × 64 = 32,768. This explains the difference in reported dimensions.
>
> Regarding evaluation on SF-XL-testv1, although NetVLAD descriptors can be precomputed offline, nearest-neighbor search requires loading over 2 million database descriptors into memory, which exceeded the hardware limits of our setup and prevented us from running the test.
>
> As for Pitts250k-test, while we acknowledge its relevance, prior works such as SALAD [1] and CricaVPR [2] often report results on only one of Pitts30k or Pitts250k, given that performance trends are consistent across both. Following this convention, we chose to evaluate on Pitts30k-test. Nevertheless, we appreciate your suggestion and will include Pitts250k-test results in the final version of the paper. Preliminary results for these methods are provided in our response to Reviewer W9pq'Q1.
>
> [1]Lzquierdo S, Civera J. Optimal transport aggregation for visual place recognition[C]/Proceedings of the IEEE/CVF Conference on Computer Vision and Pattern Recognition. 2024: 17658-17668.
> [2]Lu F, Lan X, Zhang L, et al. Cricavpr: Cross-image correlation-aware representation learning for visual place recognition[C]//Proceedings of the IEEE/CVF Conference on Computer Vision and Pattern Recognition. 2024: 16772-16782.
>
> ### Response to Q3:
> Thank you for this important question regarding our methodology for selecting the number of clusters, K.
> To clarify, we train our method only on the SF-XL dataset, using a single universal set of hyperparameters (including K, learning rate, etc.) across all experiments. Our ablation study on K (Table 4) deliberately varies both K and the backbone architecture to explore their impact on performance.
>
> Interestingly, we observe that ResNet50-based descriptors tend to produce more clearly defined cluster boundaries, while DINOv2-based features often yield overlapping clusters—in some cases, even classes with different orientation labels appear to fall within a single cluster. This indicates that ResNet50 induces more separable features, whereas DINOv2 tends to pull visually similar images closer in feature space. This observation aligns with Table 4: ResNet50 performs better with K = 6, while DINOv2 performs better with K = 3.
>
> The optimal value of K is also tied to the intrinsic structure of the training data. For example:
> - SF-XL: This dataset is ideal for our classification task, containing many multi-view images per location. These exhibit viewpoint-induced supervision inconsistencies—precisely the challenge our method is designed to address. For this dataset, K = 6 or 3 is appropriate.
> - GSV-Cities: This dataset lacks the above challenge, as all images from a location share the same orientation. Thus, K = 1 is sufficient and preferred.
>
> In summary, the optimal choice of K is related to both the characteristics of the dataset and the feature space induced by the backbone. While these trends provide useful guidance, the exact K should still be validated through empirical ablation studies.
>
> ### Response to weakness1:
> Thank you for the suggestion. We are committed to strengthening our experimental comparisons. We will include comparisons with SALAD and AnyLoc in the final manuscript. Preliminary results for these methods are provided in our response to Reviewer W9pq.
>
> ### Response to weakness2:
> EigenPlaces performs better than our method on the SF-XL-testv1 split. As noted in the main text (line 171), we attribute this to overfitting. In the EigenPlaces training pipeline, images are cropped from panoramic views facing the same direction, introducing strong viewpoint diversity. Since SF-XL-testv1 also consists mainly of multi-view test samples, this leads to overfitting.
> However, EigenPlaces does not handle occlusion well. When a fixed viewpoint is occluded by obstacles, label inconsistency arises—a core challenge our method is designed to address. This is supported by our 10-point performance gain over EigenPlaces on SF-XL-Occlusion, demonstrating EigenPlaces’ inferior generalization capability under occlusion.
>
> ### Response to weakness3:
> Regarding the selection of the number of clusters K, we have provided a detailed discussion in our response to Q3. In future work, we plan to explore different datasets and investigate how to select an optimal or dynamic K, based on dataset characteristics and the structure of the feature space.

---

> > ### Comment · Reviewer_uVwD · 2025-08-05
> >
> > Thanks for the detailed rebuttal from the authors. Most of my concerns are addressed. I would expect clear information in the revised version, including NetVLAD is retrained.

---

> > > ### Author Response · Authors · 2025-08-05
> > >
> > > We sincerely thank the reviewer for the helpful comments and for recognizing the value of our work. We will clearly describe in the revised version how we retrained NetVLAD to ensure clarity and reproducibility.

---

> > > ### Author Response · Authors · 2025-08-08
> > >
> > > We sincerely appreciate your thoughtful review and constructive comments. We hope the additional results and clarifications provided in the rebuttal help resolve your concerns and positively inform your final evaluation.

---

### Comment · Area_Chair_vs3t · 2025-08-02

Dea 19412 Reviewers: Thanks much for those who have responded to authors' detailed rebuttals!  I'd urge those who have not done that to read their rebuttals (as well as other reviews) early to allow further interactions that help clarify any lingering confusion or misunderstanding. Thank you! AC

---

### Note · Authors · 2025-08-13

We sincerely thank the reviewers for their valuable comments and constructive suggestions. In our rebuttal, we have further clarified how our method addresses the supervision inconsistency in classification-based VPR methods—arising from viewpoint changes, occlusions, and other challenging conditions—through a mutual-learning adaptive clustering approach. Specifically:

- We explained that the choice of the number of clusters K depends on both the characteristics of the dataset and the structure of the descriptor feature space.
- We clarified the reviewers’ concerns regarding the dimensionality of our retrained NetVLAD.
- We explained why our method performs slightly worse than EigenPlaces on SF-XL-test-v1: EigenPlaces is specifically tailored for extreme viewpoint changes, whereas our method strikes a balance—handling viewpoint variations while also being robust to occlusions and other extreme conditions—thus offering better generalization. We have added supporting experiments to validate this claim.
- Following the reviewers’ suggestions, we conducted additional experiments on various PEFT methods to evaluate their performance within our framework.
- Based on the reviewers’ feedback, we incorporated additional baselines (e.g., SALAD, AnyLoc, CliqueMining) and included the Pitts250k-test dataset. At comparable descriptor dimensions, our method still achieves state-of-the-art performance, and shows substantial improvement on SF-XL-Occlusion, demonstrating the novelty and effectiveness of our approach in tackling supervision inconsistency.

We will also include experiments with higher-dimensional descriptors to ensure fairer comparisons with other methods. Overall, the additional experiments, broader comparisons, and clarified explanations directly address the main concerns raised in the reviews and further strengthen the contributions of our work.

We appreciate the reviewers’ thoughtful feedback and recognition, and we hope that these substantial revisions and results will be taken into account during the final evaluation.

---

### Decision · Program_Chairs · 2025-09-17

**Decision:**

Accept (poster)

**Comment:**

The paper introduces a novel mutual learning framework for Visual Place Recognition (VPR) that eliminates the need for manual heading labels by integrating unsupervised K-means clustering with descriptor learning. Leveraging a frozen DINOv2 backbone and lightweight adapters, the method iteratively refines clustering and representation learning to overcome label inconsistency caused by viewpoint variations and occlusions. MutualVPR achieves state-of-the-art results, particularly on occlusion-heavy datasets, without relying on handcrafted rules or explicit supervision.

Reviewers praised the method’s novel unsupervised labeling strategy that aligns supervision with geographic locations and co-learns clustering and visual representations. The approach removes the need for heading labels, which are often difficult to obtain, and still shows strong improvements in VPR performance, especially on datasets with occlusions. The method maintains compact descriptors and trains only lightweight components on top of a frozen DINOv2 backbone, offering a parameter-efficient solution. The paper is described as well-written and easy to read, with clear motivation and convincing qualitative and quantitative results that demonstrate the value of the proposed framework.

Several reviewers noted missing comparisons to key recent works, including other DINOv2-based and high-performing VPR methods, which weakens the empirical positioning. The clustering process relies on a static K hyperparameter without principled selection or sensitivity analysis, and the paper lacks ablations that isolate the contributions of clustering versus the feature extractor. Some baselines were compared using lower-dimensional descriptors, which may bias the results, and certain architectural components (e.g., GeM, LMCL) were underexplained. Additional clarity in notation, more formal problem setup, and better placement of related work were also suggested to improve accessibility and rigor.

The authors provided a detailed and constructive rebuttal that addressed all major points.  They clarified design choices (e.g., descriptor dimensionality, clustering strategy), added experiments on additional datasets and methods (e.g., SALAD, AnyLoc, Pitts250k-test), and explained performance trade-offs with respect to methods like EigenPlaces. The mutual learning framework was recognized as novel and effective for handling supervision inconsistency in classification-based VPR, particularly under occlusions and viewpoint variation. With strengthened empirical results and clear revisions promised, this paper received three borderline accepts and one accept -- an acceptance consensus.